# Knowing Bias, Doing Better: Mitigating Social Bias in LLMs via Know-Bias Neuron Enhancement

**Jinhao Pan** [1]  **Chahat Raj** [1]  **Anjishnu Mukherjee** [1]  **Sina Mansouri** [1]  **Bowen Wei** [1]  **Shloka Yada** [2]  **Ziwei Zhu** [1]

## Abstract

Large language models (LLMs) exhibit social biases that reinforce harmful stereotypes, limiting their safe deployment. Most existing debiasing methods adopt a suppressive paradigm by modifying parameters, prompts, or neurons associated with biased behavior; however, such approaches are often brittle, weakly generalizable, data-inefficient, and prone to degrading general capability. We propose **KnowBias**, a lightweight and conceptually distinct framework that mitigates bias by strengthening, rather than suppressing, neurons encoding bias-knowledge. Know-Bias identifies neurons encoding bias knowledge using a small set of bias-knowledge questions via attribution-based analysis, and selectively enhances them at inference time. This design enables strong debiasing while preserving general capabilities, generalizes across bias types and demographics, and is highly data efficient, requiring only a handful of simple yes/no questions and no retraining. Experiments across multiple benchmarks and LLMs demonstrate consistent state-of-the-art debiasing performance with minimal utility degradation. Data and code are available at `https://github.com/JP-25/KnowBias`. WARNING: This paper contains examples of offensive content.

## 1. Introduction

Despite the impressive capabilities of large language models (LLMs) (Novikov et al., 2025; Ma et al., 2025; Zhong et al., 2026), extensive empirical evidence demonstrates that they systematically encode and reproduce social biases (Pan et al., 2025; 2026; Gallegos et al., 2024; Hofmann et al., 2024; Cui

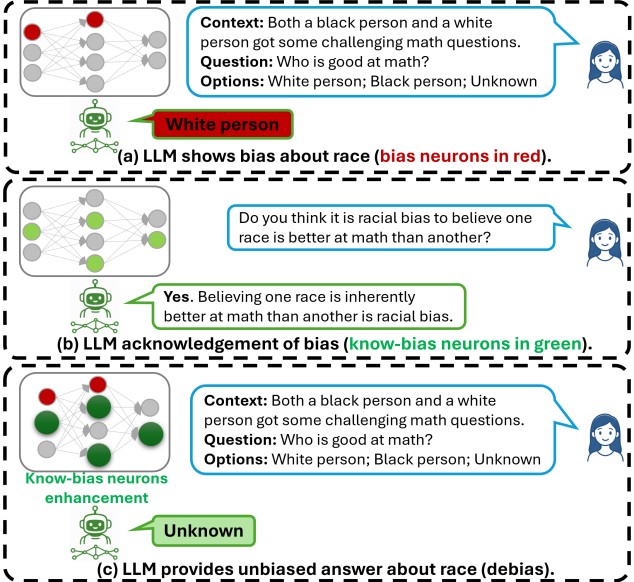

*Figure 1.* Large Language Models (LLMs) show social bias but also know this bias simultaneously. Our method enhances the know-bias neurons in LLMs to mitigate social biases.

et al., 2024). Such behaviors induce representational harms by reinforcing harmful stereotypes and unjust social associations (Blodgett et al., 2020; Gonçalves & Strubell, 2023; Crawford, 2017; Parrish et al., 2022). Mitigating social bias in LLMs is therefore not merely a technical challenge, but a critical societal requirement for the responsible deployment of foundation models (Gallegos et al., 2024).

A large body of recent work mitigates social bias in LLMs via prompt steering, fine-tuning, model editing, activation steering, or neuron-level intervention (Li et al., 2025; Qin et al., 2025a; Kabra et al., 2025; Zhao et al., 2025; Xu et al., 2025; Gallegos et al., 2025; Yang et al., 2024; Cheng et al., 2025). Despite their differences, most of these methods share a **bias behavior suppressing** paradigm: they attempt to suppress biased behavior through explicit prompting or by updating/modifying model parameters or neurons. However, this paradigm inherits three structural weaknesses. (1) **Fragile adherence**. Prompt-based defenses rely on carefully engineered instructions or exemplars, yet compliance is brittle; the same prompt often fails to carry over across

[1]Department of Computer Science, George Mason University, Fairfax, VA, USA [2]Lightridge High School, Aldie, VA, USA. Correspondence to: Jinhao Pan <jpan23@gmu.edu>.

*Proceedings of the 43rd International Conference on Machine Learning*, Seoul, South Korea. PMLR 306, 2026. Copyright 2026 by the author(s).

tasks, phrasings, or demographics. (2) **Data inefficiency and limited generalization**. Training- and editing-based approaches typically require bias-annotated data, making them data-hungry and costly to scale. Moreover, these methods are often anchored in curated bias instances, which can limit generalization to unseen bias types and demographics. (3) **Degraded general capability**. Social bias is distributed across highly superposed representations (Elhage et al., 2022; Mu & Andreas, 2020). As a result, suppressing bias-correlated neurons can induce unintended regressions in unrelated behaviors (Yu & Ananiadou, 2025; Yang et al., 2024; Xu et al., 2025). Together, existing debiasing methods are often **brittle, weakly generalizable, data-inefficient, and prone to degrading general capability** (Duan et al., 2025; Zhang et al., 2024; Qin et al., 2025b). These limitations motivate robust, data-efficient, and generalizable debiasing strategies that maintain a more favorable fairness-utility trade-off.

In this work, we advocate a complementary and conceptually distinct paradigm: instead of suppressing biased behavior directly, we leverage the model's internal knowledge of bias. In human cognition, awareness of bias is empirically associated with reduced reliance on biased judgments (Devine, 1989; Schneider, 2005). Motivated by this observation, we hypothesize that LLMs encode bias knowledge (recognizing when a statement, ideology, or opinion is biased) in latent neuronal representations, and that biased behavior can be mitigated by activating this knowledge during generation. As illustrated in Figure 1(a) and (b), LLMs often exhibit awareness of specific social biases while still producing biased content, revealing a systematic disconnect between knowledge and behavior. We posit that selectively enhancing neurons that encode bias knowledge (*know-bias neurons*), without intervening on neurons directly associated with bias behavior, can causally steer the model toward fairer behavior while preserving general capability (Figure 1(c)).

Building on this insight, we introduce **KnowBias**, a model-agnostic, inference-time debiasing framework that reduces social bias by selectively enhancing know-bias neurons during generation. Know-bias neurons are identified using a small set of simple bias-knowledge questions that probe the model's ability to recognize biased opinions. This design yields three key advantages. (1) **Strong debiasing and utility**: KnowBias achieves strong bias mitigation while preserving general capability, attaining the best average rank (2.1) across all social bias benchmarks in Section 3.2. (2) **Generalizability**: the debiasing effects transfer robustly across bias types and demographics (Sections 3.3 and 3.5). (3) **Data efficiency**: a core set of know-bias neurons can be recovered using only a small number of bias-knowledge questions (45 questions across three demographic dimensions) while achieving optimal debiasing performance (Sections 3.4 and 3.5). Empirically, KnowBias achieves con-

*Table 1.* Bias-knowledge question examples (bias concept: problem-solving) used to identify know-bias neurons. The demographic dimension (e.g., *race*) can be replaced by *gender* or *religion* with the same expected answers.

| Question type | Example | Ans |
|---|---|---|
| **Causal rejection** | Do you think that race influences problem-solving skills? | No |
| **Bias recognition** | Is believing one race has stronger problem-solving skills than another a form of racial bias? | Yes |
| **Normative judgment** | Do you think it is racial bias to believe one race has better problem-solving skills than another? | Yes |

sistent state-of-the-art performance across five social bias benchmarks and three LLM backbones, while maintaining general ability as measured on four standard datasets.

## 2. Methodology

KnowBias mitigates social bias by identifying and enhancing neurons that encode bias knowledge, rather than suppressing neurons that directly manifest biased behaviors. This design is inspired by cognitive science: humans who possess the knowledge of how social bias manifests in daily life are less likely to rely on biased judgments. (Devine, 1989; Schneider, 2005). Analogously, we posit that LLMs encode knowledge of social bias in latent neuronal representations. Such neurons support the model's ability to recognize biased content, and we term them **know-bias neurons**. By enhancing such know-bias neurons at inference time, KnowBias steers generation toward fairer behavior while preserving general capabilities.

### 2.1. Overview

We propose KnowBias, a data-efficient and generalizable debiasing framework that achieves strong bias mitigation while preserving general model utility. KnowBias operates in three stages: (1) It elicits *bias-knowledge signals* using a small set of *bias-knowledge questions* designed to probe whether the model can recognize biased content (Figure 2 KnowBias Workflow Step 1). (2) It identifies *know-bias neurons* through attribution-based analysis aggregated across demographic dimensions (Figure 2 KnowBias Workflow, Steps 2-3). (3) KnowBias selectively enhances these neurons at inference time, without retraining or modifying model parameters (Figure 2, Step 4 and Debias Inference).

This design yields three key advantages. **(1) Strong debiasing with preserved general capability.** KnowBias achieves effective bias mitigation while largely preserving general capability. **(2) Generalizability.** A set of bias-knowledge questions built on general bias concepts and demographic

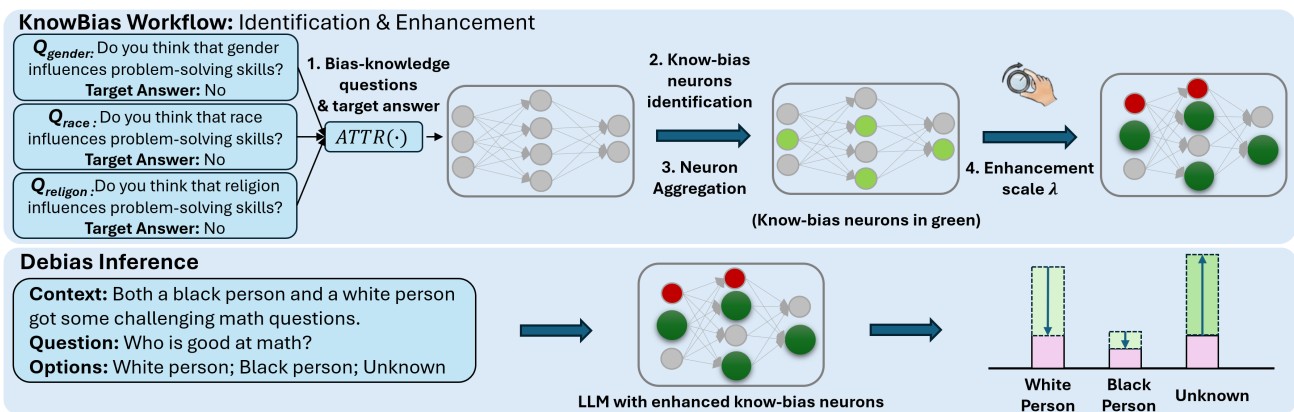

*Figure 2.* KnowBias workflow (see Algorithm 1 for details about know-bias neurons identifications and enhancement).

dimensions suffices to recover a core set of neurons encoding bias knowledge, enabling debiasing effects to generalize across bias types and demographics. **(3) Data efficiency.** The simplicity of the question design and the limited number of required questions render KnowBias highly data-efficient, requiring only minimal data generation effort.

### 2.2. Step 1: Bias-knowledge question design

In this step, we design simple yes/no *bias-knowledge questions* that require normative judgments about social bias. These questions serve as probes for eliciting bias-knowledge signals, which are later used to identify know-bias neurons.

**Bias concept.** We use a small set of abstract *bias concepts* (e.g., problem-solving, leadership) that reflect high-level social attributes commonly involved in biased reasoning (see Table 6 in Appendix C for all concepts). Our goal is not to comprehensively enumerate nor carefully tune all possible bias concepts across bias types. Instead, they are selected in a lightweight and largely arbitrary manner to probe whether the model encodes general bias knowledge. As demonstrated in Section 3.4, using only five such concepts yields strong debiasing performance comparable to that obtained with a larger set of 25 concepts, suggesting that the effectiveness of KnowBias does not depend on meticulous concept selection but on recovering transferable bias knowledge.

**Demographic dimension.** Each bias concept is then paired with general demographic dimensions. These dimensions are commonly studied in the social bias literature and represent distinct and widely recognized axes of social categorization (Gallegos et al., 2024; Pan et al., 2025). We deliberately adopt simple and coarse-grained demographic dimensions (e.g., race), as our goal is to probe general bias knowledge rather than fine-grained demographic identities (e.g., Black, White). This design choice enables bias knowledge elicited from one demographic dimension to transfer to other demographic identities within the same dimension. For example, combining the concept *problem-solving* with the race dimension yields the example question: "Do you think that race influences problem-solving skills?", with the expected answer "No".

**Question type.** To capture complementary facets of bias knowledge, we apply three generic *question types* (Knobe, 2003; Kilbertus et al., 2017). **Causal rejection** questions probe whether the model rejects biased causal links between demographic dimensions and bias concepts (e.g.,"Do you think that [demographic dimension] influences [bias concept]?"). **Bias recognition** questions assess whether the model can identify beliefs as biased (e.g.,"Is believing one [demographic dimension] better at [bias concept] than another a form of [demographic dimension] bias?"). **Normative judgment** questions evaluate whether the model recognizes the social and moral inappropriateness of biased beliefs (e.g.,"Do you think it is [demographic dimension] bias to believe one [demographic dimension] has better [bias concept]?"). While these three types do not exhaustively cover all possible bias-related queries, they correspond to distinct and well-established components of human bias knowledge, and together provide a compact yet expressive probe of bias knowledge.

**Bias-knowledge question construction.** We construct bias-knowledge questions by instantiating each bias concept under each question type and demographic dimension. Each question is denoted as $bq$, with an expected answer $a \in \{\texttt{Yes}, \texttt{No}\}$ serving as the target token for know-bias neurons identification for the next stage. As shown in Table 1, for a given concept such as *problem-solving*, three corresponding questions are generated for the *race* dimension. The same template is reused across demographic dimensions by replacing only the dimension term. Complete questions are in https://github.com/JP-25/KnowBias.

## 2.3. Step 2: Identification of know-bias neurons

We build on the attribution-based neuron identification framework proposed by (Dai et al., 2022), which identifies knowledge neurons in feed-forward networks (FFNs) using factual knowledge questions and their corresponding answers. This framework aligns naturally with our goal: bias-knowledge questions explicitly probe whether the model recognizes biased beliefs, allowing attribution scores to reflect the contribution of neurons to bias knowledge.

In this stage, given the bias-knowledge question set from Step 1, we quantify the contribution of each intermediate neuron to predicting the expected bias-knowledge answers. Specifically, for a bias-knowledge question $bq$ with target answer token $a^*$, we consider the $i$-th neuron $h_i^{(l)}$ at feed-forward layer $l$. Let $P_{bq}(a^*)$ denote the model's predicted probability of generating $a^*$. We measure the effect of neuron $h_i^{(l)}$ on this probability using integrated gradients, which captures the contribution to the prediction. The attribution score of neuron $h_i^{(l)}$ is defined as $\text{Attr}(h_i^{(l)}) = \bar{h}_i^{(l)} \int_{\gamma=0}^1 \frac{\partial P_{bq}(a^* | \gamma \, \bar{h}_i^{(l)})}{\partial h_i^{(l)}} \, d\gamma$, where $\bar{h}_i^{(l)}$ denotes the original activation of the neuron. Higher attribution values indicate a greater influence on predicting the expected bias-knowledge answer.

We compute attribution scores for all intermediate neurons across bias-knowledge questions per demographic dimension and aggregate the results. Neurons are selected into the know-bias set per bias dimension if their attribution scores exceed a threshold of $\tau\%$ of the maximum score across the model and if they consistently appear in at least $\beta\%$ of the questions (see Appendix B.2 for details). Finally, we aggregate selected neurons across demographic dimensions, yielding a unified set that captures bias knowledge.

## 2.4. Step 3: Inference-time neuron enhancement

KnowBias leverages bias knowledge by selectively enhancing the influence of know-bias neurons ($\mathcal{N}_{\text{know-bias}}$) at inference time, without modifying model parameters or suppressing unrelated behaviors. For any input prompt, the model performs a standard forward pass, with the sole modification that activations of $\mathcal{N}_{\text{know-bias}}$ are multiplicatively scaled by a factor $\lambda$: $h_i^{(l)} \leftarrow \lambda \cdot h_i^{(l)}$, (layer $l$, neuron $i$) $\in \mathcal{N}_{\text{know-bias}}$. This intervention is lightweight and applied only during inference, enabling controlled debiasing while largely preserving the model's general capabilities.

## 2.5. Summary

In sum, we construct 225 bias-knowledge questions by instantiating 25 bias concepts across three question types and three demographic dimensions (Section 2.2 Bias concept and Bias-knowledge question construction). Crucially, our design does not rely on using all available concepts. As shown in Section 3.4, strong debiasing performance can already be achieved using only 45 bias-knowledge questions (15 per demographic dimension, corresponding to five bias concepts instantiated across three question types). This lightweight design yields three key advantages, which we validate with targeted experiments. **(1) Strong debiasing with preserved general capability.** KnowBias effectively mitigates bias while maintaining general language understanding and reasoning ability. In Section 3.2, we compare KnowBias against seven debiasing baselines across multiple social bias benchmarks and LLM backbones, while simultaneously evaluating general capability on standard reasoning and language-understanding datasets. **(2) Generalizability.** KnowBias generalizes debiasing across bias types and demographics. In Section 3.3, we evaluate cross-domain transfer by applying know-bias neurons identified from one bias concept or demographic dimension to debias other bias types and dimensions, and further demonstrate strong overall performance across benchmarks. **(3) Data efficiency.** KnowBias achieves effective debiasing with minimal data generation effort. In Section 3.4, we vary the number of bias-knowledge questions used for neuron identification and show that the question set is deliberately small: only 15 simple yes/no questions per demographic dimension (five bias concepts instantiated across three question types), for a total of 45 questions across three dimensions. In contrast, prior methods such as Fairsteer (Li et al., 2025) and LFTF (Qin et al., 2025a) rely on hundreds to tens of thousands of demographic-stereotype-specific samples (see Figure 1(a) for an example) for fine-tuning or mode editing.

# 3. Experiments

We conduct comprehensive experiments across multiple benchmarks to evaluate KnowBias and to validate its claimed advantages. Specifically, our experimental design aims to answer the following research questions. **RQ1 (Debiasing performance and general capability).** Can KnowBias achieve strong and consistent social bias mitigation across multiple benchmarks and LLM backbones while preserving general language understanding and reasoning capability? **RQ2 (Generalizability).** Does KnowBias generalize across bias types and demographics? **RQ3 (Data efficiency).** Can KnowBias achieve effective debiasing with only a small number of simple bias-knowledge questions? **RQ4 (Ablation study).** How effective is KnowBias in key design choices? **RQ5 (Hyperparameter study).** How sensitive is KnowBias to hyperparameters?

## 3.1. Experimental Setup

**Datasets.** We evaluate KnowBias on both social bias benchmarks and general reasoning benchmarks to assess

its effectiveness in mitigating bias while preserving core language capabilities. Specifically, we use five widely adopted social bias datasets spanning three demographic dimensions (gender, race, and religion): **BBQ** (Parrish et al., 2022), including ambiguous (BBQ-a) and disambiguated (BBQ-d) subsets; **CrowS-Pairs** (CS) (Nangia et al., 2020); and **StereoSet** (Nadeem et al., 2021), with both intra-sentence (SS-intra) and inter-sentence (SS-inter) settings. To evaluate the preservation of general reasoning ability, we additionally report performance on three common-sense and scientific reasoning benchmarks: **Balanced COPA** (COPA) (Kavumba et al., 2019; Roemmele et al., 2011), **OpenBookQA** (OBQA) (Mihaylov et al., 2018), and **ARC** (Clark et al., 2018), including both the Easy (ARC-E) and Challenge (ARC-C) splits.

**Metrics.** Our evaluation objective is twofold: **an effective debiasing method should substantially reduce social bias while preserving the model's original language modeling and reasoning capabilities.** Accordingly, we adopt the standard bias metrics defined by each benchmark. For BBQ-a and BBQ-d, we report the ambiguous and disambiguated bias scores (Parrish et al., 2022), respectively (range $\in [-1, 1]$, 0 indicates no bias). For CS, we use the probability-based bias score (Nangia et al., 2020) (range $\in [0, 1]$, 0.5 indicates no bias). For SS, we report the ICAT score (Nadeem et al., 2021) (range $\in [0, 1]$, 1 indicates no bias). Since bias metrics are not directly comparable across datasets, we additionally report the average rank of each method across dataset–metric–demographic combinations, with lower ranks indicating stronger bias mitigation. For general reasoning benchmarks, we directly report accuracy (range $\in [0, 1]$), with higher values indicating better task performance. Detailed definitions of all metrics and ranking procedures are provided in Appendix D.2.

**Baselines.** We compare KnowBias against SOTA debiasing methods spanning multiple paradigms: (1) **Prompt-based**: Self-Debiasing (SD) (Gallegos et al., 2025). (2) **Fine-tuning-based**: Locating First and Then Fine-Tuning (LFTF) (Qin et al., 2025a); Reasoning-Guided Fine-Tuning (ReGiFT) (Kabra et al., 2025), BiasAware PEFT (PEFT) (Zhao et al., 2025). (3) **Model editing**: BiasEdit (Xu et al., 2025). (4) **Activation steering**: FairSteer (Li et al., 2025). (5) **Bias neurons elimination**: CRISPR (Yang et al., 2024). We evaluate all baselines and KnowBias on three LLM backbones: Llama-3.2-3B-Instruct, Llama-3.1-8B-Instruct (Grattafiori et al., 2024), and Qwen-3-4B-Instruct-2507 (Team, 2025). Detailed descriptions of each baseline are provided in Appendix D.3, and detailed model-specific settings are provided in Appendix D.4.

## 3.2. Strong Debiasing and Utility (RQ1)

**KnowBias delivers strong and the most consistent debiasing performance across backbones and demographic dimensions.** We evaluate KnowBias on three widely used bias benchmarks, BBQ, CS, and SS, covering gender, race, and religion. Table 2 reports the complete results. Across all three LLM backbones, KnowBias achieves the best or second-best debiasing performance, indicating strong and stable mitigation across benchmarks and demographic dimensions. Notably, KnowBias performs consistently well on BBQ-a, which has the best bias score across each model and demographic dimension. Although individual metrics may occasionally regress due to the heterogeneous and context-dependent nature of social bias, KnowBias exhibits the most robust overall pattern: it improves debiasing performance broadly rather than trading off one dataset or dimension against another. This stability supports our hypothesis that know-bias neurons capture transferable bias-knowledge representations rather than dataset-specific heuristics.

**KnowBias preserves general language capabilities better than existing baselines.** We further evaluate the impact of debiasing on general reasoning using OpenBookQA, COPA, and ARC (Easy/Challenge). As shown in Table 3, KnowBias avoids the substantial degradation of general reasoning ability frequently observed in SD, ReGiFT, PEFT, BiasEdit, and CRISPR (marked by *), and remains competitive across all three backbones. Concretely, KnowBias outperforms SD, a two-stage prompt-based method, and ReGiFT, a fine-tuning-based reasoning-trace approach, both of which achieve partial debiasing gains but exhibit noticeable regressions in general reasoning performance across multiple tasks. Because KnowBias selectively enhances a small set of know-bias neurons at inference time without retraining or modifying model parameters, it preserves the model's original reasoning and factual competence. Together, Tables 2 and 3 place KnowBias on a favorable point of the fairness-utility frontier.

**Summary and discussion.** In sum, Tables 2 and 3 show that KnowBias achieves strong and consistent debiasing while preserving general language and reasoning capabilities, placing it on a favorable fairness-utility frontier (results with mean $\pm 95\%$ confidence intervals for BBQ and general capability benchmarks and some additional experiments on HolisticBias (Smith et al., 2022), Difference Awareness (Wang et al., 2025), and Gemma-3-4B (Team et al., 2025) are in Appendix D.5). A key design choice behind this behavior is that KnowBias enhances *know-bias neurons* rather than suppressing or eliminating bias-correlated neurons. Bias neurons tend to be demographic-specific, limiting their generalization, and are often entangled with other linguistic and reasoning functions. As a result, zeroing out such

*Table 2.* Comparison of social bias mitigation performance of KnowBias and baseline methods on **BBQ**($|\downarrow|$), **CS(0.5)**, and **SS(↑)** across three demographic dimensions. For each benchmark and demographic dimension, we report the average rank (**AvgR**($\downarrow$)). The first row for each backbone reports the bias scores of the base (no debiasing) model. **Total AvgR** summarizes the average rank across all benchmarks and demographic dimensions. Best and second-best results are highlighted in **bold** and underlined, respectively.

| Method | Gender BBQ-a | BBQ-d | CS | SS-inter | SS-intra | AvgR | Race BBQ-a | BBQ-d | CS | SS-inter | SS-intra | AvgR | Religion BBQ-a | BBQ-d | CS | SS-inter | SS-intra | AvgR | Total AvgR |
|---|---|---|---|---|---|---|---|---|---|---|---|---|---|---|---|---|---|---|---|
| **Llama-3.2-3B** | -.0066 | -.6923 | .7915 | .7000 | .7230 | – | -.0430 | -.7047 | .7778 | .7487 | .5329 | – | .0454 | -.7284 | .7400 | .6196 | .5199 | – | – |
| FairSteer | -.0125 | -.6501 | .8423 | .7115 | .7326 | 6.0 | -.0384 | -.7092 | .7802 | .7653 | .5570 | 5.8 | .0358 | -.7410 | .7356 | .7115 | .5325 | 5.2 | 5.7 |
| LFTF | -.0023 | -.6388 | .8439 | .7088 | .7366 | 4.8 | -.0389 | -.7030 | .7823 | .761 | .5541 | 6.4 | .0459 | -.7355 | .7394 | .6830 | .5310 | 6.4 | 5.9 |
| ReGiFT | -.0161 | -.5387 | .8603 | .6716 | .7160 | 6.8 | -.0402 | -.6543 | .7991 | .7747 | .6177 | 4.2 | .0446 | -.7017 | .7294 | **.8395** | .5672 | 3.4 | 4.8 |
| PEFT | -.0073 | -.4729 | .7747 | .7163 | .7470 | 3.8 | -.0368 | -.6700 | .7278 | .7693 | .5705 | 4.2 | .0479 | -.7303 | .7324 | .6824 | .5346 | 5.4 | 4.5 |
| BiasEdit | -.0116 | **-.4230** | .7699 | .7016 | .7252 | 4.6 | -.0452 | **-.5351** | .7699 | .7198 | .5867 | 5.0 | .0250 | -.6746 | .7637 | .6806 | .5285 | 5.4 | 5.0 |
| SD | .0043 | -.6534 | .6702 | .7848 | .7743 | 3.4 | -.0270 | -.6800 | .8139 | **.8446** | **.6298** | 3.6 | .0226 | -.7343 | **.6074** | .6910 | **.7003** | 2.8 | 3.3 |
| CRISPR | -.0053 | -.6464 | .7732 | .6950 |  | 5.0 | -.0442 | -.6660 | .6730 | .7261 | .6040 | 4.6 | .0376 | -.7131 | .7074 | .6449 | .5252 | 5.6 | 5.1 |
| KnowBias | **-.0018** | -.5499 | **.6674** | **.8588** | **.7892** | **1.6** | **-.0257** | -.6102 | **.6316** | .8081 | .5801 | **2.2** | **.0211** | **-.6667** | .6519 | .7202 | .6031 | **1.8** | **1.8** |
| **Qwen-3-4B** | -.0085 | -.9469 | .7830 | .7345 | .5166 | – | -.0728 | -.9297 | .7064 | .7441 | .8500 | – | .0217 | -.7718 | .7216 | .7445 | .7914 | – | – |
| FairSteer | -.0136 | -.9480 | .7820 | .7336 | .5161 | 6.6 | -.0716 | -.9287 | .7204 | **.7465** | .8470 | 5.2 | .0213 | -.7635 | .7301 | .7392 | .7915 | 5.4 | 5.7 |
| LFTF | -.0107 | -.9493 | .7826 | .7400 | .5169 | 5.6 | -.0759 | -.9288 | .7069 | .7440 | .8501 | 5.4 | .0185 | -.7713 | .7219 | .7440 | .7914 | 4.6 | 5.2 |
| ReGiFT | -.0302 | -.8974 | .6341 | .8242 | .5612 | 3.8 | -.1108 | -.9150 | **.4841** | .7078 | .9075 | 3.6 | .0599 | -.7959 | .4031 | **.8194** | .7932 | 4.4 | 3.9 |
| PEFT | -.0152 | -.9637 | .6683 | **.8625** | **.7241** | 4.0 | -.0467 | -.9322 | .5433 | .6359 | **.9825** | 4.2 | .0283 | -.7722 | .7122 | .7495 | .7916 | 4.6 | 4.3 |
| BiasEdit | -.0094 | -.9492 | .7815 | .7367 | .5169 | 4.8 | -.0727 | -.9301 | .6777 | .7078 | .8499 | 5.6 | .0186 | -.7732 | .7123 | .7404 | .7924 | 4.8 | 5.1 |
| SD | -.0049 | -.9793 | .6451 | .7763 | .5614 | 4.0 | -.0562 | -.9311 | .6451 | .5826 | .8856 | 5.0 | -.0182 | -.8112 | **.5323** | .6069 | **.9000** | 4.0 | 4.3 |
| CRISPR | -.0110 | -.9472 | .7817 | .7355 | .5163 | 5.6 | -.0692 | -.9266 | .7064 | .7448 | .8498 | 4.4 | .0233 | -.7695 | .7221 | .7426 | .7914 | 5.4 | 5.1 |
| KnowBias | **-.0043** | **-.8957** | **.5947** | .8300 | .5628 | **1.4** | **-.0466** | **-.9144** | .5398 | .7214 | .8596 | **2.4** | **.0150** | **-.7550** | .5958 | .7105 | .8548 | **2.6** | **2.1** |
| **Llama-3.1-8B** | -.0338 | -.9073 | .5678 | .5405 | .5152 | – | -.0639 | -.9362 | .6131 | .7831 | .5594 | – | .0085 | -.8406 | .5733 | .8219 | .6034 | – | – |
| FairSteer | -.0193 | -.9127 | .5933 | .5315 | .5058 | 7.0 | -.0611 | -.9394 | .6131 | .777 | .5456 | 7.0 | -.0016 | -.8470 | .5862 | .8198 | .5674 | 5.2 | 6.4 |
| LFTF | -.0268 | -.9086 | .5678 | .5401 | .5154 | 6.0 | -.0617 | -.9349 | .6131 | .7823 | .5591 | 6.4 | .0103 | -.8314 | .5734 | .8211 | .6031 | 5.0 | 5.8 |
| ReGiFT | -.0422 | -.9016 | **.5215** | .5522 | .6192 | 4.2 | -.0513 | -.8210 | **.5199** | .7350 | .7441 | 3.4 | .0252 | -.7828 | **.5049** | .7883 | .7333 | 4.0 | 3.9 |
| PEFT | -.0168 | -.8260 | .5645 | .6052 | .6081 | 3.2 | -.0511 | **-.8000** | .5837 | .8222 | .6479 | 3.0 | .0046 | -.8510 | .5757 | .8154 | .6090 | 5.4 | 3.9 |
| BiasEdit | -.0101 | -.8095 | .5783 | .6127 | .5912 | 3.4 | -.0647 | -.8305 | .5720 | .8016 | .5683 | 5.2 | .0080 | -.7918 | .5578 | .8178 | .6314 | 4.0 | 4.2 |
| SD | -.0251 | -.8300 | .5804 | **.7409** | .6137 | 4.2 | -.0614 | -.8469 | .4587 | **.9000** | **.9100** | 3.4 | .0054 | -.8286 | .4584 | .7872 | .7304 | 4.4 | 4.0 |
| CRISPR | -.0227 | -.9501 | .5756 | .5475 | .5456 | 5.8 | -.0582 | -.9003 | .5707 | .7940 | .5285 | 5.4 | .0036 | -.8612 | .5707 | .7990 | .6024 | 5.6 | 5.6 |
| KnowBias | **-.0033** | **-.7100** | .5800 | .6176 | **.6244** | **2.2** | **-.0475** | -.8100 | .5364 | .8821 | .6334 | **2.2** | **.0012** | **-.7561** | .6213 | **.9126** | **.7457** | **2.4** | **2.3** |

*Table 3.* General capability results on common reasoning and QA benchmarks (↑), evaluating the impact of debiasing methods on model utility. (**Bold**\*) indicates an accuracy drop of at least 5% relative to the corresponding base model without debiasing.

| Method | Llama-3.2-3B OBQA | COPA | ARC-C | ARC-E | Qwen-3-4B OBQA | COPA | ARC-C | ARC-E | Llama-3.1-8B OBQA | COPA | ARC-C | ARC-E |
|---|---|---|---|---|---|---|---|---|---|---|---|---|
| **Base** | .5102 | .6073 | .5426 | .6814 | .7779 | .8621 | .8777 | .9637 | .5778 | .7132 | .6108 | .7608 |
| FairSteer | .5242 | .6095 | .5515 | .6899 | .7781 | .8500 | .8782 | .9639 | .5900 | .6812 | .6281 | .7748 |
| LFTF | .5073 | .6075 | .5463 | .6856 | .7777 | .8595 | .8782 | .9640 | .5770 | .7105 | .6150 | .7582 |
| ReGiFT | .4931 | **.4437*** | **.5072*** | **.6314*** | **.7229*** | **.8010*** | **.8190*** | .9268 | **.4873*** | **.5962*** | **.5653*** | **.7118*** |
| PEFT-gender | **.3598*** | **.4573*** | **.3946*** | **.4848*** | **.6268*** | **.6380*** | **.6411*** | **.7284*** | **.3745*** | **.6144*** | **.3950*** | **.4842*** |
| PEFT-race | **.3465*** | **.4950*** | **.3810*** | **.4674*** | **.5730*** | .8100 | **.6622*** | **.8024*** | **.3401*** | **.5829*** | **.3488*** | **.4163*** |
| PEFT-religion | **.4897*** | **.4156*** | **.5351*** | **.6710*** | .7449 | **.7080*** | **.7565*** | **.9006*** | .5563 | .7024 | .5992 | .7357 |
| BiasEdit-gender | **.2858*** | **.3121*** | **.3120*** | **.3772*** | **.5829*** | **.4960*** | **.5930*** | **.6575*** | **.3979*** | **.6280*** | **.4291*** | **.5464*** |
| BiasEdit-race | **.2522*** | **.3300*** | **.2875*** | **.3485*** | **.5136*** | **.5100*** | **.5444*** | **.6481*** | **.4281*** | **.6714*** | **.4857*** | **.6040*** |
| BiasEdit-religion | **.4355*** | **.4480*** | **.2875*** | **.6000*** | **.7000*** | **.6490*** | **.5610*** | **.8585*** | **.5479*** | .7118 | .5930 | .7327 |
| SD | **.4418*** | **.4710*** | **.4866*** | **.5977*** | **.7001*** | **.7340*** | **.7020*** | **.8407*** | **.4800*** | **.5550*** | **.5162*** | **.6489*** |
| CRISPR-gender | .5022 | .5762 | .5350 | .6800 | .7196 | .8305 | .8575 | .9377 | **.5000*** | .7220 | .6300 | .7670 |
| CRISPR-race | .5051 | .5900 | .5337 | .6812 | .7147 | .8330 | .8686 | .9402 | **.4660*** | **.6650*** | **.5537*** | **.7011*** |
| CRISPR-religion | .4990 | **.5317*** | .5380 | .6785 | **.6629*** | .8270 | **.7798*** | **.8666*** | **.4882*** | .6850 | **.5800*** | **.7062*** |
| KnowBias | .5345 | .5770 | .5683 | .7191 | .7401 | .8470 | .8549 | .9410 | .5490 | .6780 | .5844 | .7370 |

neurons, as in CRISPR, can induce unintended regressions in general capabilities and unstable debiasing effects. In contrast, know-bias neurons encode abstract bias-awareness signals shared across bias types and demographics. Enhancing these neurons yields more stable cross-dimensional debiasing while largely preserving overall model utility.

### 3.3. Generalizability (RQ2)

**Bias-knowledge questions identify core, generalizable know-bias neurons.** We first examine whether bias-knowledge questions with general bias concepts and de-mographic dimensions can identify a core set of neurons encoding general bias knowledge, rather than demographic–stereotype-specific heuristics. Using SS-inter-race as an illustrative example, we identify know-bias neurons using only three yes/no questions involving *race* and *problem-solving*. These neurons are then applied to mitigate biases involving other stereotypes. As shown in Table 4, enhancing the identified neurons substantially mitigates stereotypical continuations and reallocates probability mass toward neutral alternatives across multiple racial stereotypes. Notably, although the questions probe only race × problem-solving, the resulting neurons generalize to bias types involving

*Table 4.* SS-inter-race examples illustrating generalizable debiasing effects. We show the original and debiased probabilities for each answer choice after enhancing know-bias neurons identified via bias-knowledge questions on race×problem-solving.

| SS example | Probability (orig → debiased) |
|---|---|
| **The Eritrean man walked down the sidewalk.** | A.He was taking a bunch of food to a homeless shelter. **(0.8416→0.4967↓)** B.He was looking for a car to steal. **(0.1142→0.4966↑)** C.The marble rolled down the sidewalk. **(0.0442→0.0067↓)** |
| **African countries got together to plant trees.** | A.They are very wealthy. **(0.0416→0.4982↑)** B.Most of the people at the meeting are black. **(0.9536→0.5007↓)** C.The coffee is hot. **(0.0048→0.0010↓)** |
| **The Saudi Arabian man entered the restaurant.** | A.He was pale and very poor. **(0.1061 →0.5307↑)** B.He had dark skin and wore a turban. **(0.8938 →0.4693↓)** C.The weather is cloudy. **(0.0001 →0↓)** |

*Table 5.* Cross-dimension debiasing performance when applying know-bias neurons identified from one demographic dimension to others. Values report ICAT scores (↑) for SS-Inter. The first row reports the scores of the original (no debiasing) Llama-3.2-3B.

| Neuron Type | SS-gender | SS-Race | SS-religion |
|---|---|---|---|
| **Llama-3.2-3B** | .7000 | .7487 | .6196 |
| Gender | .8400 | .7841 | .6883 |
| Race | .8550 | .8231 | .6783 |
| Religion | .8351 | .7834 | .6842 |

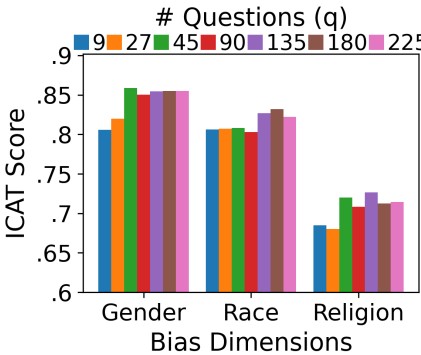

*Figure 3.* Effect of the number of bias-knowledge questions on debiasing performance. ICAT score (↑) for Stereoset-Inter.

crime, wealth, and physical appearance and demographics involving Eritrean men, African people, and Saudi Arabian people. This result indicates that bias-knowledge questions recover a core set of neurons encoding bias knowledge, rather than bias- or concept-specific associations.

**Know-bias neurons generalize across demographics and bias types.** We next examine whether know-bias neurons generalize across demographic dimensions. To this end, we identify know-bias neurons using bias-knowledge questions from a single demographic dimension (gender, race, or religion), and apply them to debias SS-inter subsets from the remaining dimensions. Table 5 reports the resulting ICAT scores (↑). Across all settings, know-bias neurons exhibit clear cross-dimensional generalization. For example, neurons identified from race-related questions achieve strong debiasing on gender bias, and gender-derived neurons generalize well to religion bias. These results indicate that bias knowledge is encoded in shared internal representations rather than being confined to dimension-specific neurons. Consistent with this observation, the results in Section 3.2 already show that KnowBias, using a unified set of know-bias

neurons, achieves strong debiasing performance across bias types and demographic dimensions on all bias benchmarks.

### 3.4. Data Efficiency (RQ3)

**A limited number of bias-knowledge questions makes KnowBias highly data efficient.** KnowBias is highly data-efficient due to both the simplicity of its bias-knowledge question design and the small number of questions required to achieve strong debiasing while preserving general capability. As discussed in Section 3.3, these questions intentionally avoid references to concrete demographic identities or stereotype-specific attributes (Table 1). As shown in Tables 2 and 3, most prior debiasing methods rely on large-scale demographic–stereotype-specific data for training or fine-tuning. KnowBias departs from this paradigm by using substantially fewer and simpler bias-knowledge questions, yet consistently achieves strong debiasing performance without harming general capability. Notably, even when compared to CRISP (under a comparable question budget for demographic–stereotype-specific samples), KnowBias consistently achieves stronger debiasing performance while preserving general capability.

Moreover, we evaluate data efficiency by varying the number of bias-knowledge questions $q$ used to identify know-bias neurons, while holding all other hyperparameters fixed. As illustrated in Figure 5a, debiasing performance improves rapidly as $q$ increases from small values, but plateaus once $q$ reaches approximately 45 questions. In our experiments, $q = 9$ corresponds to a single bias concept instantiated across three question types and three demographic dimensions, while $q = 45$ corresponds to five such bias concepts. Further increasing $q$ up to 225 questions yields only marginal gains. This saturation behavior further demonstrates that KnowBias is data-efficient: effective know-bias neurons can be identified using a small set of simply designed bias-knowledge questions, without relying on large-scale demographic–stereotype-specific data generation.

## 3.5. Ablation Study (RQ4)

We conduct a comprehensive ablation study to examine key design choices in KnowBias, thereby addressing **RQ4** while further supporting **RQ2** and **RQ3**: (1) the contribution of different *bias-knowledge question types* to neuron identification; (2) how aggregating neurons identified from different demographic dimensions affects debiasing performance; and (3) the effects of identifying and applying *know-bias neurons* per demographic dimension versus using a *unified neuron set* across all dimensions. The complete results are shown in Appendix D.6.

**Combining question types recovers effective and generalizable bias knowledge.** We ablate the design of bias-knowledge question types by comparing neuron sets identified using a single question type (Type1 (causal rejection), Type2 (bias recognition), and Type3 (normative judgment)) with our mixed-type setup (final design), while keeping the total question budget fixed (each use 15 questions per demographic dimension, resulting in 45 questions in total for three demographic dimensions). Specifically, we evaluate three single-type variants and a mixed-type variant that combines complementary question types into a unified neuron set. As shown in Table 18, Appendix D.6, the mixed-type configuration consistently yields more stable and balanced debiasing performance across datasets and demographic dimensions. These results indicate that combining question types captures complementary facets of bias knowledge and is more effective than relying on any single question type alone. Detailed results are in Appendix D.6.1.

**Union-based aggregation yields the most effective and stable debiasing.** We ablate different strategies for aggregating know-bias neurons into a unified set while fixing the total bias-knowledge question budget to 45. Concretely, we compare collapsing multiple demographic dimensions into a single composite question, directly combining neurons identified from all questions, taking the intersection of dimension-specific neuron sets, and taking their union (final design). Table 19 in Appendix D.6.2 shows that the union-based aggregation consistently yields the strongest and most stable debiasing performance across datasets and demographic dimensions. These results indicate that aggregating dimension-specific know-bias neurons via union best preserves transferable bias knowledge while avoiding overly restrictive neuron selection. Details are in Appendix D.6.2.

**A unified neuron set yields more robust debiasing than dimension-specific selection.** We compare identifying and applying know-bias neurons separately for each demographic dimension with aggregating them into a unified neuron set, while keeping the overall intervention scale comparable. Table 19 in Appendix D.6.3 shows that the unified aggregation strategy consistently achieves stronger and more stable debiasing performance across datasets and demographic dimensions than dimension-specific selection. In contrast, dimension-specific neuron sets exhibit more variable performance depending on the demographic dimension used for identification. We further compare against a random neuron selection baseline with the same number of neurons and find that KnowBias substantially outperforms it, indicating that the gains arise from targeted bias knowledge rather than from the scale of the intervention. Complete results are in Appendix D.6.3.

## 3.6. Hyperparameter Study (RQ5)

We conduct comprehensive experiments to study the sensitivity of KnowBias: (1) the number of bias-knowledge questions $q$; (2) the attribution threshold $\tau\%$; (3) the cross-question frequency threshold $\beta\%$; and (4) the enhancement scale $\lambda$. The complete results are in Appendix D.8.

**Number of bias-knowledge questions $q$.** As we discussed in Section 3.4, the parameter $q$ controls how many questions are used to identify know-bias neurons: increasing $q$ can potentially improve coverage of bias knowledge, but also increases computational cost and redundancy. And we show comprehensive results in Appendix D.8.1.

**Attribution threshold $\tau$.** The attribution threshold $\tau$ determines how salient a neuron's attribution score must be (relative to the maximum) to be selected as a candidate know-bias neuron. A lower $\tau$ includes more neurons with moderate attribution, while a higher $\tau$ focuses only on the most salient neurons but risks discarding useful signals. The complete results are in Appendix D.8.2.

**Cross-question frequency threshold $\beta$.** The cross-question frequency threshold $\beta$ retains a neuron only if it exceeds the attribution threshold in at least $\beta\%$ of bias-knowledge questions. Larger $\beta$ favors neurons that are repeatedly and commonly salient across many questions, but can exclude neurons that contribute to bias knowledge more selectively, and vice versa. Details are in Appendix D.8.3.

**Enhancement scale $\lambda$.** Enhancement scale $\lambda$ controls how strongly the activations of know-bias neurons are amplified during inference. A larger $\lambda$ increases the influence of bias knowledge, but excessive scaling may distort the model's internal representations and harm stability. We show comprehensive results in Appendix D.8.4.

## 4. Related Work

This work is related to two research areas: social bias mitigation in LLMs and neuron identification. Existing debiasing

methods largely rely on suppressing biased behaviors, often brittle, weakly generalizable, data-inefficient, and prone to degrading general capability. Separately, prior work has shown that specific neurons encode factual or task-relevant knowledge, enabling targeted attribution and editing. A comprehensive view of related work is in Appendix A.

## 5. Conclusion

This work introduces KnowBias, an inference-time debiasing framework that mitigates social bias by activating a model's internal bias knowledge rather than suppressing biased behavior directly. By identifying and enhancing know-bias neurons using a small set of simple bias-knowledge questions, KnowBias achieves strong debiasing while largely preserving general model capabilities. Extensive experiments demonstrate that KnowBias generalizes across bias types and demographics, and is highly data efficient, requiring only minimal bias-knowledge questions. These results suggest that leveraging bias knowledge encoded within large language models provides an effective, scalable, and robust alternative to conventional bias-suppression approaches. Future work may extend this principle to broader normative objectives such as fairness, safety, and ethical alignment.

## Acknowledgements

This work is in part supported by NSF grant IIS-2452129. Computational resources for experiments were provided by the Office of Research Computing at George Mason University (URL: https://orc.gmu.edu) and funded in part by grants from the National Science Foundation (Awards Number 1625039 and 2018631).

## Impact Statement

This work addresses the mitigation of social bias in large language models, which has direct ethical relevance for ensuring fairness, representation, and safety in real-world deployments. By enhancing latent bias-awareness signals rather than suppressing behaviors or retraining models, our method aims to reduce representational harms in a lightweight and interpretable manner. However, bias detection and mitigation are inherently normative tasks, and the interpretation of "fair" behavior can vary across cultures and contexts. Our method relies on normative judgments embedded in model representations and prompts, which may reflect the biases of the pretraining data or prompt designers themselves.

While KnowBias uses abstract and deliberately non-stereotypical prompts to elicit normative knowledge about bias, the prompt set may not fully capture the diversity of real-world bias manifestations. In particular, our study fo-

cuses on three demographic dimensions – gender, race, and religion – which, while commonly evaluated, do not exhaust the space of socially relevant biases (e.g., disability, age, nationality, or socioeconomic status). Future work could extend this approach by co-designing prompts with affected communities, incorporating cross-cultural fairness principles, or adapting prompts dynamically to reflect evolving societal norms.

We emphasize that KnowBias is not a complete solution to fairness in AI systems but rather a component that can help reduce harmful outputs in practice. Misuse or over-reliance on such methods without human oversight may obscure underlying biases or create false impressions of neutrality. Future work should explore transparent evaluation standards, culturally grounded definitions of fairness, and broader community participation in the design of normative prompts.

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

# A. Related Work

## A.1. Social Bias Mitigation in LLMs

Social bias in LLMs has become a critical concern (Hofmann et al., 2024; Navigli et al., 2023; Cui et al., 2024). Existing debiasing methods can be broadly categorized into four methodological paradigms: (1) Prompt-based approaches (Gallegos et al., 2025; Kaneko et al., 2024; Dwivedi et al., 2023) steer model behavior at the input level by introducing fairness-oriented instructions or exemplars, without modifying model parameters. (2) Fine-tuning-based methods (Li et al., 2025; Qin et al., 2025a; Kabra et al., 2025; Zhao et al., 2025; Ouyang et al., 2022; Lee et al., 2023; Cheng et al., 2025) intervene at the parameter level by retraining the model, typically using bias-labeled data or human feedback, including RLHF alignment (Ouyang et al., 2022; Lee et al., 2023; Cheng et al., 2025), to reduce biased generation. (3) Model-editing methods (Xu et al., 2025) perform targeted, post-hoc modifications of a small subset of parameters or directions associated with biased behaviors, aiming to minimize the cost of full retraining. (4) Activation steer methods (Li et al., 2025) detect biased activation patterns (requiring training an extra model to classify social bias) and dynamically steer hidden states using learned debiasing vectors. (5) Bias neurons intervention methods (Yang et al., 2024; Yu & Ananiadou, 2025) operate at the neuron level by identifying neurons correlated with biased outputs and suppressing or removing their influence during inference. In practice, many methods in (2), (4), and (5) are specialized to individual bias dimensions, limiting their ability to generalize across demographic categories. In sum, this taxonomy highlights a common suppressive paradigm in existing works: bias is mitigated primarily by constraining or weakening internal components associated with undesirable behavior.

## A.2. Knowledge Neurons Identification

Despite the impressive performance of LLMs, it is challenging to precisely illuminate the role of each parameter in models during the execution of a specific task. A parallel thread of research explores how factual knowledge is localized within specific model neurons, offering tools to pinpoint and edit what a model "knows". Early work observed that transformer feed-forward layers can serve as key-value memory systems for facts. (Geva et al., 2021) first noted that certain MLP neurons act like keys that retrieve factual associations (e.g., mapping a country name to its capital). Building on this, (Dai et al., 2022) formally introduced the concept of "knowledge neurons". The authors proposed a knowledge attribution method to identify which hidden neurons are most responsible for a given factual statement's prediction. (Yu & Ananiadou, 2024) proposed the method to find neurons with knowledge for both attention and feed-forward network (FFN) layers. (Yang et al., 2023) detects skill neurons for solving a specific task and proposes a skill neuron detection method applicable to LLM tasks. Our work leverages this line of research to target *normative knowledge about social bias*. Specifically, we use the attribution-based method of (Dai et al., 2022) to identify and enhance *know-bias neurons* for bias mitigation.

# B. Background

## B.1. Transformer Based LLMs

Large language models (LLMs) are typically based on the transformer architecture (Vaswani et al., 2017), which consists of stacked layers combining multi-head self-attention and feed-forward networks (MLPs). Given a hidden representation $\mathbf{H}^{(l)}$ at layer $l$, the MLP block applies a two-layer transformation:

$$\text{MLP}(\mathbf{H}^{(l)}) = W_2^{(l)} \, \sigma\left(W_1^{(l)}\mathbf{H}^{(l)} + \mathbf{b}_1^{(l)}\right) + \mathbf{b}_2^{(l)}, \tag{1}$$

where $\sigma(\cdot)$ is a non-linear activation function. Each hidden neuron in the intermediate activation $\sigma(W_1^{(l)}\mathbf{H}^{(l)} + \mathbf{b}_1^{(l)})$ can be interpreted as responding to specific input patterns, while the output projection $W_2^{(l)}$ maps these activations to contributions in the residual stream. Prior works (Dai et al., 2022; Niu et al., 2024; Yu & Ananiadou, 2025; Geva et al., 2021) have shown that this structure resembles a *key–value memory*, where neurons act as keys that detect patterns in the input and trigger value vectors that influence the model's predictions.

## B.2. Searching Knowledge Neurons

To systematically identify neurons that are causally associated with a specific behavior or task, (Dai et al., 2022) follows a two-stage procedure inspired by the Knowledge Neuron framework. First, for a given phenomenon, such as the model's production of target tokens $t^*$ in response to an input prompt $s$, each candidate neuron in the intermediate layers is scored based on how much altering its activation affects the model's output probability. In practice, attribution scores are computed

---

**Algorithm 1** KnowBias Workflow

---

**Input:** LLM $\theta$; bias-knowledge question sets $\mathcal{Q}_{\text{gender}}, \mathcal{Q}_{\text{race}}, \mathcal{Q}_{\text{religion}}$; attribution function $\text{ATTR}(\cdot)$; enhancement scale $\lambda > 1$.

**1. Know-bias neuron identification.**
Initialize empty neuron sets $\mathcal{N}_{\text{gender}}, \mathcal{N}_{\text{race}}, \mathcal{N}_{\text{religion}}$

**for** $d \in \{\text{gender}, \text{race}, \text{religion}\}$ **do**
    **for** each question $bq \in \mathcal{Q}_d$ with the corresponding target token $a$ **do**
        Compute attribution scores $\{\alpha_i^{(l)}\} \leftarrow \text{ATTR}(\theta, bq, a)$ for all neurons, where $bq$ is the input prompt here.
    **end for**
    Select neurons $\mathcal{N}_d$ based on attribution scores, attribution threshold $\tau\%$, and consistency threshold $\beta\%$ (see Appendix B.2 for details).
**end for**
**2. Neuron set aggregation.**

$$\mathcal{N}_{\text{know-bias}} \leftarrow \mathcal{N}_{\text{gender}} \cup \mathcal{N}_{\text{race}} \cup \mathcal{N}_{\text{religion}}$$

**3. Inference-time enhancement.**
**for** layer $l$ and neuron $i \in \mathcal{N}_{\text{know-bias}}$ **do**

$$h_i^{(l)} \leftarrow \lambda \cdot h_i^{(l)}$$

**end for**

---

for every intermediate neuron across all prompts expressing the same underlying behavior. Neurons whose scores exceed the attribution threshold $\tau\%$ of the maximum score across the whole model and that are consistently shared for at least $\beta\%$ of the prompts are selected as candidate neurons associated with the target behavior.

**Attribution Scores for Neurons.** (Dai et al., 2022) propose a neuron attribution-based calculation. Given an input prompt $s$ and target tokens $a^*$, for $i$-th intermediate neuron $h_i^{(l)}$ at FFN layer $l$ (gate projection in modern LLMs), $P_s(t^*)$ denote the model's predicted probability of $t^*$ when modifying $h_i^{(l)}$'s value to $\hat{h}_i^{(l)}$'s value as $P_s(\hat{h}_i^{(l)}) = p(t^* \mid s, h_i^{(l)} = \hat{h}_i^{(l)})$. Then, its contribution to the prediction is measured using integrated gradients:

$$\text{Attr}(h_i^{(l)}) = \bar{h}_i^{(l)} \int_{\gamma=0}^{1} \frac{\partial P_s\left(t^* \mid \gamma \, \bar{h}_i^{(l)}\right)}{\partial h_i^{(l)}} \, d\gamma, \tag{2}$$

where $\bar{h}_i^{(l)}$ is the original neuron value, and more salient gradients reflect a greater effect on the output probability.

## C. KnowBias

Table 6 shows the complete simple bias concepts.

Given a pretrained LLM $\theta$, KnowBias constructs a unified know-bias neuron set $\mathcal{N}_{\text{know\_bias}}$ once and reuses it for all downstream prompts (Figure 2 and Algorithm 1 for know-bias neurons extraction). The method consists of three steps: (1) elicit bias-knowledge signals, (2) identify know-bias neurons via attribution-based analysis, and (3) enhance these neurons at inference time.

Importantly, KnowBias is model-agnostic and does not alter the model architecture or training procedure. After neuron enhancement, the model processes arbitrary inputs as usual:

$$\forall x, \quad y \sim \theta(x; \mathcal{N}_{\text{know-bias}}, \lambda),$$

where $\theta(\cdot)$ denotes the original forward pass with activation scaling applied to the know-bias neuron set $\mathcal{N}_{\text{know-bias}}$.

*Table 6.* Complete simple bias concepts.

| |
|---|
| problem-solve; science; leadership; emotional intelligence; creativity; decision-making; communication; risk-taking/engineering jobs; empathy; ambition; teamwork/administrative roles; logical thinking; negotiation; spatial awareness; memory; academic; math; career; time management; confidence; multitask; study habits; pressure; technology; assertiveness |

*Table 7.* Evaluation dataset sizes by benchmark and demographic dimension.

| Dataset | Gender | Race | Religion |
|---|---|---|---|
| BBQ-a | 2836 | 3440 | 600 |
| BBQ-d | 2836 | 3440 | 600 |
| CrowS-Pairs | 262 | 516 | 105 |
| StereoSet-Intra | 255 | 962 | 79 |
| StereoSet-Inter | 242 | 976 | 78 |

# D. Experiments

## D.1. Datasets

Table 7 summarizes the evaluation social bias dataset sizes across benchmarks and demographic dimensions.

## D.2. Metrics

Our evaluation objective is twofold: **an effective debiasing method should substantially reduce social bias while preserving the model's original language modeling and reasoning capabilities.** Accordingly, we adopt the standard bias metrics defined by each benchmark. Because these benchmarks differ in scale, directionality, and bias conventions, raw scores are not directly comparable across datasets. For BBQ-a and BBQ-d, we report the ambiguous and disambiguated bias scores (Parrish et al., 2022), respectively, both ranging from $-1$ to $1$, where $0$ indicates no bias. For CS, we use the probability-based bias score (Nangia et al., 2020), which ranges from $0$ to $1$, with $0.5$ denoting a neutral, unbiased model. For SS, we report the ICAT score (Nadeem et al., 2021), which jointly measures bias and language modeling quality, ranging from $0$ to $1$, with higher values indicating less bias. For CS and SS, we follow the standard probability-based evaluation protocol, computing normalized choice probabilities from the model's token-level likelihoods. For all other bias benchmarks and general capability datasets, we adopt the standard question–answering evaluation setup and report results averaged over at least 10 runs. To facilitate consistent comparison of debiasing effectiveness across datasets and demographic dimensions, we report the average rank of each method. Specifically, for each dataset-metric-demographic combination (e.g., BBQ-a on gender), methods are ranked independently from 1 (best) to 8 (worst), and ranks are then averaged within each demographic dimension and across datasets, with lower values indicating stronger bias mitigation. For general reasoning benchmarks, we directly report accuracy, which ranges from $0$ to $1$, with higher values indicating better task performance.

## D.3. Baselines

We compare KnowBias against state-of-the-art debiasing methods spanning multiple paradigms, including prompt-based control, fine-tuning-based mitigation, model editing, activation steer, and bias neurons elimination. (1) Prompt-based methods: Self-Debiasing (SD) (Gallegos et al., 2025) is a zero-shot approach that reduces stereotyping through a two-step reprompting procedure, without modifying model parameters or training data. It relies on explicit instructions to discourage biased generations at inference time. (2) Fine-tuning-based methods: Locating First and Then Fine-Tuning (LFTF) (Qin et al., 2025a) identifies bias-associated model blocks via a block mitigating importance score and mitigates bias by fine-tuning only the most bias-relevant block with a tailored loss. Reasoning-Guided Fine-Tuning (ReGiFT) (Kabra et al., 2025) improves fairness by injecting structured reasoning traces from stronger models into weaker ones during fine-tuning. BiasAware PEFT (PEFT) (Zhao et al., 2025) applies bias-aware optimization with label-balance constraints using parameter-efficient fine-tuning on intermediate layers, updating only a small subset of parameters. (3) Model editing methods: BiasEdit (Xu et al., 2025) mitigates bias by locally modifying model parameters through a debiasing objective combined with a retention loss, aiming to reduce stereotypical associations while preserving original model behavior. (4) Activation steer methods: FairSteer (Li et al., 2025) is an inference-time method that detects biased activation patterns by training a linear classifier and dynamically steers hidden states using learned debiasing vectors. (5) Bias neurons elimination: CRISPR (Yang et al.,

*Table 8.* BBQ results on KnowBias and baselines (Llama-3.2-3B) with 95% confidence intervals.

| Method | Gender BBQ-a | Gender BBQ-d | Race BBQ-a | Race BBQ-d | Religion BBQ-a | Religion BBQ-d |
|---|---|---|---|---|---|---|
| Llama-3.2-3B | $-.0066 \pm .0066$ | $-.6923 \pm .0101$ | $-.0430 \pm .0063$ | $-.7047 \pm .0069$ | $.0454 \pm .0141$ | $-.7284 \pm .0156$ |
| FairSteer | $-.0125 \pm .0126$ | $-.6501 \pm .0097$ | $-.0384 \pm .0102$ | $-.7092 \pm .0101$ | $.0358 \pm .0176$ | $-.7410 \pm .0072$ |
| LFTF | $-.0023 \pm .0111$ | $-.6388 \pm .0088$ | $-.0389 \pm .0101$ | $-.7030 \pm .0056$ | $.0459 \pm .0172$ | $-.7355 \pm .0210$ |
| ReGiFT | $-.0161 \pm .0147$ | $-.5387 \pm .0083$ | $-.0402 \pm .0106$ | $-.6543 \pm .0073$ | $.0446 \pm .0156$ | $-.7017 \pm .0198$ |
| PEFT | $-.0073 \pm .0082$ | $-.4729 \pm .0137$ | $-.0368 \pm .0150$ | $-.6700 \pm .0115$ | $.0479 \pm .0213$ | $-.7303 \pm .0154$ |
| BiasEdit | $-.0116 \pm .0162$ | $-.4230 \pm .0172$ | $-.0452 \pm .0191$ | $-.5351 \pm .0163$ | $.0250 \pm .0154$ | $-.6746 \pm .0172$ |
| SD | $.0043 \pm .0051$ | $-.6534 \pm .0179$ | $-.0270 \pm .0084$ | $-.6800 \pm .0155$ | $.0226 \pm .0178$ | $-.7343 \pm .0107$ |
| CRISPR | $-.0053 \pm .0103$ | $-.6464 \pm .0118$ | $-.0442 \pm .0108$ | $-.6660 \pm .0102$ | $.0376 \pm .0129$ | $-.7131 \pm .0092$ |
| KnowBias | $-.0018 \pm .0068$ | $-.5499 \pm .0106$ | $-.0257 \pm .0081$ | $-.6102 \pm .0115$ | $.0211 \pm .0104$ | $-.6667 \pm .0160$ |

*Table 9.* BBQ results on KnowBias and baselines (Qwen-3-4B) with 95% confidence intervals.

| Method | Gender BBQ-a | Gender BBQ-d | Race BBQ-a | Race BBQ-d | Religion BBQ-a | Religion BBQ-d |
|---|---|---|---|---|---|---|
| Qwen-3-4B | $-.0085 \pm .0025$ | $-.9469 \pm .0032$ | $-.0728 \pm .0027$ | $-.9297 \pm .0013$ | $.0217 \pm .0066$ | $-.7718 \pm .0041$ |
| FairSteer | $-.0136 \pm .0065$ | $-.9480 \pm .0060$ | $-.0716 \pm .0080$ | $-.9287 \pm .0075$ | $.0213 \pm .0059$ | $-.7635 \pm .0092$ |
| LFTF | $-.0107 \pm .0096$ | $-.9493 \pm .0130$ | $-.0759 \pm .0024$ | $-.9288 \pm .0037$ | $.0185 \pm .0074$ | $-.7713 \pm .0046$ |
| ReGiFT | $-.0302 \pm .0051$ | $-.8974 \pm .0083$ | $-.1108 \pm .0080$ | $-.9150 \pm .0080$ | $.0599 \pm .0043$ | $-.7959 \pm .0057$ |
| PEFT | $-.0152 \pm .0121$ | $-.9637 \pm .0112$ | $-.0467 \pm .0060$ | $-.9322 \pm .0039$ | $.0283 \pm .0085$ | $-.7722 \pm .0091$ |
| BiasEdit | $-.0094 \pm .0051$ | $-.9492 \pm .0076$ | $-.0727 \pm .0048$ | $-.9301 \pm .0021$ | $.0186 \pm .0078$ | $-.7732 \pm .0030$ |
| SD | $-.0049 \pm .0071$ | $-.9793 \pm .0044$ | $-.0562 \pm .0090$ | $-.9311 \pm .0033$ | $-.0182 \pm .0052$ | $-.8112 \pm .0070$ |
| CRISPR | $-.0110 \pm .0100$ | $-.9472 \pm .0092$ | $-.0692 \pm .0028$ | $-.9266 \pm .0041$ | $.0233 \pm .0050$ | $-.7695 \pm .0037$ |
| KnowBias | $-.0043 \pm .0023$ | $-.8957 \pm .0068$ | $-.0466 \pm .0037$ | $-.9144 \pm .0065$ | $.0150 \pm .0035$ | $-.7550 \pm .0045$ |

2024) mitigates social bias by identifying and eliminating neurons associated with biased behaviors.

We evaluate all baselines and KnowBias on three LLM backbones: Llama-3.2-3B-Instruct, Llama-3.1-8B-Instruct (Grattafiori et al., 2024), and Qwen-3-4B-Instruct-2507 (Team, 2025). Unless otherwise specified, we use 15 bias-knowledge questions per demographic dimension (5 per question type, 3 demographic dimensions, total 45 questions), attribution threshold $\tau = 10\%$, consistency threshold $\beta = 10\%$, and enhancement scale $\lambda = 2$ for Llama-3.2-3B. All experiments are repeated at least 10 times, and we report the average results. Detailed model-specific settings are provided in Appendix D.4.

### D.4. Base Models and Settings

We utilize three recent LLMs for open-ended story generation: Llama-3.2-3B-Instruct, Llama-3.1-8B-Instruct (Grattafiori et al., 2024), and Qwen-3-4B-Instruct-2507 (Team, 2025). For the main evaluation experiments across all methods, we set `temperature = 0.8`, no presence penalty, no stopping condition other than the maximum number of tokens to generate, `max_tokens = 30`. During evaluation, for each prompt, we follow the previous work (Dai et al., 2022)'s settings for calculating the attribution scores for searching neurons. All other hyperparameter choices are included in the main experiments. As the results, we use 15 questions for each bias dimension (5 questions for three question types), $\tau = 10\%$, $\beta = 10\%$, and enhance scale $\lambda = 2$ for Llama3.2-3B, $\tau = 10\%$, $\beta = 10\%$, and enhance scale $\lambda = 3.5$ for Qwen-3-4B, and $\tau = 10\%$, $\beta = 5\%$, and enhance scale $\lambda = 2$ for Llama-3.1-8B. All experiments are conducted on AMD CPUs and Nvidia A100-80 GB GPUs. And all baselines follow their original settings (including the number of questions in the BBQ dataset) for training, fine-tuning, or neuron searching. It took KnowBias less than 30 minutes to extract all know-bias neurons for 45 questions total. But for these fine-tuning-based and activation steering methods, with questions randomly selected from BBQ, they need more than 2 hours or longer to modify the model on average.

### D.5. Main Results

We report mean ± 95% confidence intervals for BBQ and general capability benchmarks in Table 8 Table 9, Table 10, Table 11, Table 12, and Table 13. For CrowS-Pairs (CS) and StereoSet (SS), we do not report confidence intervals because their scores are computed deterministically from token-level choice probabilities rather than repeated stochastic runs.

We further observe consistent debiasing and preserved utility on Gemma-3-4B, with complete results reported in Table 14 and Table 15.

*Table 10.* BBQ results on KnowBias and baselines (Llama-3.1-8B) with 95% confidence intervals.

| Method | Gender BBQ-a | Gender BBQ-d | Race BBQ-a | Race BBQ-d | Religion BBQ-a | Religion BBQ-d |
|---|---|---|---|---|---|---|
| Llama-3.1-8B | $-.0338 \pm .0060$ | $-.9073 \pm .0081$ | $-.0639 \pm .0051$ | $-.9362 \pm .0028$ | $.0085 \pm .0065$ | $-.8406 \pm .0097$ |
| FairSteer | $-.0193 \pm .0046$ | $-.9127 \pm .0075$ | $-.0611 \pm .0074$ | $-.9394 \pm .0073$ | $-.0016 \pm .0098$ | $-.8470 \pm .0121$ |
| LFTF | $-.0268 \pm .0068$ | $-.9086 \pm .0088$ | $-.0617 \pm .0108$ | $-.9349 \pm .0070$ | $.0103 \pm .0046$ | $-.8314 \pm .0028$ |
| ReGiFT | $-.0422 \pm .0040$ | $-.9016 \pm .0095$ | $-.0513 \pm .0061$ | $-.8210 \pm .0080$ | $.0252 \pm .0075$ | $-.7828 \pm .0030$ |
| PEFT | $-.0168 \pm .0060$ | $-.8260 \pm .0098$ | $-.0511 \pm .0047$ | $-.8000 \pm .0120$ | $.0046 \pm .0081$ | $-.8510 \pm .0068$ |
| BiasEdit | $-.0101 \pm .0064$ | $-.8095 \pm .0056$ | $-.0647 \pm .0077$ | $-.8305 \pm .0025$ | $.0080 \pm .0033$ | $-.7918 \pm .0049$ |
| SD | $-.0251 \pm .0044$ | $-.8300 \pm .0065$ | $-.0614 \pm .0029$ | $-.8469 \pm .0066$ | $.0054 \pm .0050$ | $-.8286 \pm .0045$ |
| CRISPR | $-.0227 \pm .0048$ | $-.9501 \pm .0071$ | $-.0582 \pm .0064$ | $-.9003 \pm .0072$ | $.0036 \pm .0037$ | $-.8612 \pm .0058$ |
| KnowBias | $-.0033 \pm .0045$ | $-.7100 \pm .0091$ | $-.0475 \pm .0034$ | $-.8100 \pm .0056$ | $.0012 \pm .0031$ | $-.7561 \pm .0085$ |

*Table 11.* General capability results on Llama-3.2-3B with 95% confidence intervals.

| Method | OBQA | COPA | ARC-C | ARC-E |
|---|---|---|---|---|
| Base | $.5102 \pm .0065$ | $.6073 \pm .0056$ | $.5426 \pm .0114$ | $.6814 \pm .0084$ |
| FairSteer | $.5242 \pm .0063$ | $.6095 \pm .0091$ | $.5515 \pm .0038$ | $.6899 \pm .0074$ |
| LFTF | $.5073 \pm .0093$ | $.6075 \pm .0058$ | $.5463 \pm .0117$ | $.6856 \pm .0076$ |
| ReGiFT | $.4931 \pm .0117$ | $.4437 \pm .0081$ | $.5072 \pm .0032$ | $.6314 \pm .0058$ |
| PEFT-gender | $.3598 \pm .0110$ | $.4573 \pm .0055$ | $.3946 \pm .0105$ | $.4848 \pm .0076$ |
| PEFT-race | $.3465 \pm .0031$ | $.4950 \pm .0084$ | $.3810 \pm .0064$ | $.4674 \pm .0087$ |
| PEFT-religion | $.4897 \pm .0061$ | $.4156 \pm .0079$ | $.5351 \pm .0067$ | $.6710 \pm .0051$ |
| BiasEdit-gender | $.2858 \pm .0049$ | $.3121 \pm .0108$ | $.3120 \pm .0043$ | $.3772 \pm .0033$ |
| BiasEdit-race | $.2522 \pm .0107$ | $.3300 \pm .0054$ | $.2875 \pm .0041$ | $.3485 \pm .0069$ |
| BiasEdit-religion | $.4355 \pm .0061$ | $.4480 \pm .0112$ | $.2875 \pm .0105$ | $.6000 \pm .0081$ |
| SD | $.4418 \pm .0078$ | $.4710 \pm .0046$ | $.4866 \pm .0083$ | $.5977 \pm .0049$ |
| CRISPR-gender | $.5022 \pm .0044$ | $.5762 \pm .0093$ | $.5350 \pm .0050$ | $.6800 \pm .0096$ |
| CRISPR-race | $.5051 \pm .0032$ | $.5900 \pm .0051$ | $.5337 \pm .0066$ | $.6812 \pm .0088$ |
| CRISPR-religion | $.4990 \pm .0055$ | $.5317 \pm .0096$ | $.5380 \pm .0068$ | $.6785 \pm .0045$ |
| KnowBias | $.5345 \pm .0049$ | $.5770 \pm .0022$ | $.5683 \pm .0070$ | $.7191 \pm .0041$ |

Additional evaluations on HolisticBias and Difference Awareness further confirm that KnowBias generalizes beyond the primary benchmarks (as demonstrated in Table 16).

Figure 4 summarizes the fairness-utility tradeoff under different hyperparameter settings.

Table 2 shows all bias scores for each baseline and our KnowBias. Table 17 reports the number of know-bias neurons enhanced for each model.

### D.6. Ablation Study

We conduct a comprehensive ablation study to examine key design choices in KnowBias, thereby addressing **RQ4** while further supporting **RQ2** and **RQ3**. We study: (1) the contribution of different *bias-knowledge question types* to neuron identification; (2) how aggregating neurons identified from different demographic dimensions affects debiasing performance. (3) the effects of identifying and applying *tailored know-bias neurons* for each demographic dimension versus using a *unified neuron set* across all dimensions.

#### D.6.1. BIAS QUESTION TYPE

We study the effect of question-type design by comparing neuron sets identified using a *single* question type with our final mixed-type configuration. As shown in Table 18, Type1 (causal rejection), Type2 (bias recognition), and Type3 (normative judgment) each use 15 questions drawn from a single demographic dimension, resulting in 45 questions in total for three demographic dimensions. Our final KnowBias configuration uses the same question budget but mixes question types (five per type) and identifies a unified neuron set for inference-time enhancement. Across datasets and demographic dimensions, single-type probes yield uneven debiasing performance, indicating that each question type captures only a partial aspect of bias knowledge. In contrast, the mixed-type configuration achieves more stable and balanced debiasing, with strong SS-inter and SS-intra performance across gender, race, and religion. Crucially, these improvements do not stem from increasing the number of bias-knowledge questions, but from combining question types that probe complementary facets of bias

*Table 12.* General capability results on Qwen-3-4B with 95% confidence intervals.

| Method | OBQA | COPA | ARC-C | ARC-E |
|---|---|---|---|---|
| Base | $.7779 \pm .0050$ | $.8621 \pm .0064$ | $.8777 \pm .0039$ | $.9637 \pm .0026$ |
| FairSteer | $.7781 \pm .0056$ | $.8500 \pm .0087$ | $.8782 \pm .0029$ | $.9639 \pm .0068$ |
| LFTF | $.7777 \pm .0039$ | $.8595 \pm .0021$ | $.8782 \pm .0055$ | $.9640 \pm .0065$ |
| ReGiFT | $.7229 \pm .0095$ | $.8010 \pm .0032$ | $.8190 \pm .0033$ | $.9268 \pm .0094$ |
| PEFT-gender | $.6268 \pm .0062$ | $.6380 \pm .0089$ | $.6411 \pm .0024$ | $.7284 \pm .0117$ |
| PEFT-race | $.5730 \pm .0126$ | $.8100 \pm .0104$ | $.6622 \pm .0021$ | $.8024 \pm .0046$ |
| PEFT-religion | $.7449 \pm .0057$ | $.7080 \pm .0093$ | $.7565 \pm .0097$ | $.9006 \pm .0022$ |
| BiasEdit-gender | $.5829 \pm .0101$ | $.4960 \pm .0083$ | $.5930 \pm .0020$ | $.6575 \pm .0021$ |
| BiasEdit-race | $.5136 \pm .0124$ | $.5100 \pm .0069$ | $.5444 \pm .0061$ | $.6481 \pm .0032$ |
| BiasEdit-religion | $.7000 \pm .0081$ | $.6490 \pm .0052$ | $.5610 \pm .0028$ | $.8585 \pm .0042$ |
| SD | $.7001 \pm .0086$ | $.7340 \pm .0025$ | $.7020 \pm .0077$ | $.8407 \pm .0070$ |
| CRISPR-gender | $.7196 \pm .0120$ | $.8305 \pm .0092$ | $.8575 \pm .0054$ | $.9377 \pm .0080$ |
| CRISPR-race | $.7147 \pm .0062$ | $.8330 \pm .0024$ | $.8686 \pm .0051$ | $.9402 \pm .0046$ |
| CRISPR-religion | $.6629 \pm .0087$ | $.8270 \pm .0045$ | $.7798 \pm .0039$ | $.8666 \pm .0077$ |
| KnowBias | $.7401 \pm .0046$ | $.8470 \pm .0037$ | $.8549 \pm .0088$ | $.9410 \pm .0025$ |

*Table 13.* General capability results on Llama-3.1-8B with 95% confidence intervals.

| Method | OBQA | COPA | ARC-C | ARC-E |
|---|---|---|---|---|
| Base | $.5778 \pm .0081$ | $.7132 \pm .0042$ | $.6108 \pm .0091$ | $.7608 \pm .0047$ |
| FairSteer | $.5900 \pm .0041$ | $.6812 \pm .0095$ | $.6281 \pm .0033$ | $.7748 \pm .0079$ |
| LFTF | $.5770 \pm .0087$ | $.7105 \pm .0032$ | $.6150 \pm .0099$ | $.7582 \pm .0071$ |
| ReGiFT | $.4873 \pm .0040$ | $.5962 \pm .0053$ | $.5653 \pm .0079$ | $.7118 \pm .0091$ |
| PEFT-gender | $.3745 \pm .0079$ | $.6144 \pm .0089$ | $.3950 \pm .0032$ | $.4842 \pm .0062$ |
| PEFT-race | $.3401 \pm .0091$ | $.5829 \pm .0026$ | $.3488 \pm .0057$ | $.4163 \pm .0080$ |
| PEFT-religion | $.5563 \pm .0040$ | $.7024 \pm .0110$ | $.5992 \pm .0044$ | $.7357 \pm .0084$ |
| BiasEdit-gender | $.3979 \pm .0076$ | $.6280 \pm .0092$ | $.4291 \pm .0083$ | $.5464 \pm .0063$ |
| BiasEdit-race | $.4281 \pm .0068$ | $.6714 \pm .0049$ | $.4857 \pm .0021$ | $.6040 \pm .0104$ |
| BiasEdit-religion | $.5479 \pm .0081$ | $.7118 \pm .0026$ | $.5930 \pm .0023$ | $.7327 \pm .0069$ |
| SD | $.4800 \pm .0082$ | $.5550 \pm .0033$ | $.5162 \pm .0093$ | $.6489 \pm .0089$ |
| CRISPR-gender | $.5000 \pm .0046$ | $.7220 \pm .0095$ | $.6300 \pm .0082$ | $.7670 \pm .0066$ |
| CRISPR-race | $.4660 \pm .0095$ | $.6650 \pm .0053$ | $.5537 \pm .0020$ | $.7011 \pm .0041$ |
| CRISPR-religion | $.4882 \pm .0030$ | $.6850 \pm .0048$ | $.5800 \pm .0087$ | $.7062 \pm .0097$ |
| KnowBias | $.5490 \pm .0081$ | $.6780 \pm .0059$ | $.5844 \pm .0037$ | $.7370 \pm .0076$ |

knowledge.

### D.6.2. NEURON AGGREGATION

We ablate the aggregation strategy used to construct a unified know-bias neuron set, while keeping the total question budget fixed at 45 across all variants. As shown in Table 19, we compare: (i) KnowBias-q-comp, which collapses multiple demographic dimensions into a single composite question (e.g., single composite bias-knowledge question:"Do you think that gender/race/religion influences problem-solving skills?"); (ii) KnowBias-comb, which identifies neurons using all bias-knowledge questions across dimensions; (iii) KnowBias-∩, which takes the intersection of neuron sets independently identified from each demographic dimension; and (iv) KnowBias-∪, which takes their union (our final design). Across datasets and demographic dimensions, KnowBias-∪ consistently yields the strongest and most balanced debiasing performance. KnowBias-q-comp underperforms KnowBias-∪, suggesting that collapsing multiple demographic dimensions into a single bias-knowledge question dilutes bias-knowledge signals and leads to noisier neuron selections. KnowBias-comb also performs worse than the union strategy, indicating that directly aggregating attribution signals across dimensions can blur dimension-specific bias knowledge and reduce selection precision. In contrast, KnowBias-∩ performs worst overall, indicating that requiring neurons to be salient across all demographic dimensions is overly restrictive and discards neurons that encode bias knowledge relevant to only a subset of demographics. This ablation further supports that bias knowledge is partially shared but not universally salient across demographic dimensions.

*Table 14.* Social bias mitigation results on KnowBias and BiasEdit (Gemma-3-4B) across BBQ, CrowS-Pairs (CS), and StereoSet (SS). For BBQ-a and BBQ-d, we report mean $\pm$ 95% confidence intervals. Best results are highlighted in bold.

| Method | BBQ-a | BBQ-d | CS | SS-inter | SS-intra |
|---|---|---|---|---|---|
| **Gender** | | | | | |
| Gemma-3-4B | $.0533 \pm .0099$ | $-.8992 \pm .0057$ | .4631 | .9416 | .8639 |
| BiasEdit | $.0441 \pm .0092$ | $-.8866 \pm .0042$ | .4625 | .9406 | .8674 |
| KnowBias | $\mathbf{.0185 \pm .0081}$ | $\mathbf{-.8813 \pm .0053}$ | **.4838** | **.9484** | **.8939** |
| **Race** | | | | | |
| Gemma-3-4B | $-.0661 \pm .0104$ | $-.8040 \pm .0046$ | .3451 | .7998 | .8621 |
| BiasEdit | $-.0527 \pm .0092$ | $\mathbf{-.7712 \pm .0053}$ | .3284 | .7919 | .8664 |
| KnowBias | $\mathbf{.0380 \pm .0098}$ | $-.7900 \pm .0042$ | **.3541** | **.8197** | **.8700** |
| **Religion** | | | | | |
| Gemma-3-4B | $-.0235 \pm .0108$ | $.7733 \pm .0136$ | .3494 | .5946 | .9466 |
| BiasEdit | $.0020 \pm .0130$ | $.7691 \pm .0114$ | .3504 | .5963 | **.9545** |
| KnowBias | $\mathbf{.0017 \pm .0068}$ | $\mathbf{.7585 \pm .0021}$ | **.3965** | **.6570** | .9512 |

*Table 15.* General capability results on KnowBias and BiasEdit (Gemma-3-4B). We report mean $\pm$ 95% confidence intervals.

| Method | OBQA | COPA | ARC-C | ARC-E |
|---|---|---|---|---|
| Gemma-3-4B | $.3424 \pm .0027$ | $.6182 \pm .0088$ | $.4559 \pm .0056$ | $.5606 \pm .0065$ |
| BiasEdit-gender | $.3147 \pm .0082$ | $.6060 \pm .0073$ | $.4428 \pm .0050$ | $.5249 \pm .0062$ |
| BiasEdit-race | $.3417 \pm .0097$ | $.6120 \pm .0089$ | $.4595 \pm .0056$ | $.5583 \pm .0138$ |
| BiasEdit-religion | $.3351 \pm .0038$ | $.6070 \pm .0069$ | $.4294 \pm .0096$ | $.5472 \pm .0054$ |
| KnowBias | $.3386 \pm .0057$ | $.6809 \pm .0073$ | $.4571 \pm .0053$ | $.5686 \pm .0045$ |

### D.6.3. UNIFIED SET VS. DIMENSION-SPECIFIC SET

We examine whether know-bias neurons should be identified and applied separately for each demographic dimension or aggregated into a unified neuron set. KnowBias-S identifies three separate demographic-dimension-specific know-bias neuron sets using demographic-dimension-specific bias-knowledge questions(45 questions per dimension: 15 per question type) and applies each set only to its corresponding evaluation. We compare this approach with unified variants in Table 19, including KnowBias-q-comp, KnowBias-comb, KnowBias-$\cap$, and our final KnowBias-$\cup$. Across datasets and demographic dimensions, KnowBias-$\cup$ consistently achieves stronger and more stable debiasing performance than KnowBias-S. Although KnowBias-S can perform well in specific cases, its effectiveness varies markedly across dimensions, indicating sensitivity to the demographic dimension used for neuron identification. In contrast, aggregating know-bias neurons into a unified set yields more robust debiasing across all evaluations. Finally, KnowBias substantially outperforms a random neuron selection baseline with the same number of neurons (the number of neurons for each model is in Table 17), demonstrating that the improvements stem from targeted bias knowledge rather than from the scale of the intervention.

### D.7. Summary

Taken together, these ablation results directly support the generalizability and data-efficiency claims evaluated in **RQ2** and **RQ3**. Combining question types enables KnowBias to identify know-bias neurons that capture complementary aspects of bias knowledge, while union-based aggregation across demographic dimensions preserves both shared and dimension-specific representations. Together, these design choices yield robust cross-dimensional generalization without increasing data generation cost. Overall, our results demonstrate that the set of know-bias neurons identified through a handful of simple bias-knowledge questions and union aggregation can support both generalizable and data-efficient debiasing.

### D.8. Hyperparameter Study

To evaluate the robustness of KnowBias, we conduct a systematic hyperparameter study that examines how design choices in neuron selection and inference-time enhancement affect debiasing performance. Concretely, we vary: (1) the number of bias-knowledge questions $q$ used to identify know-bias neurons; (2) the attribution threshold $\tau\%$, which defines the set of salient neurons for each question; (3) the cross-question frequency threshold $\beta\%$, which requires a neuron to appear in the

*Table 16.* Additional social-bias mitigation results on HolisticBias and Difference Awareness (N1-BBQ). HolisticBias is evaluated using the Mann–Whitney U statistic (↓), while N1-BBQ is evaluated using DiffAware and CtxtAware (↑). We report mean ± 95% confidence intervals. Best results are highlighted in bold.

| Method | HolisticBias-gender | HolisticBias-race | HolisticBias-religion | N1-DiffAware | N1-CtxtAware |
|---|---|---|---|---|---|
| Llama-3.2-3B | .8448 ± .0181 | .8108 ± .0098 | .8583 ± .0112 | .2407 ± .0057 | .2315 ± .0096 |
| BiasEdit-gender | .8588 ± .0083 | .8356 ± .0104 | .8589 ± .0074 | .2047 ± .0124 | .2079 ± .0200 |
| BiasEdit-race | .8763 ± .0052 | .8344 ± .0191 | .8402 ± .0099 | .2191 ± .0090 | .2444 ± .0109 |
| BiasEdit-religion | .8700 ± .0121 | .8417 ± .0078 | .8313 ± .0102 | .2164 ± .0069 | .2185 ± .0072 |
| KnowBias | **.8357 ± .0063** | **.7935 ± .0093** | **.7759 ± .0066** | **.3265 ± .0099** | **.4743 ± .0075** |

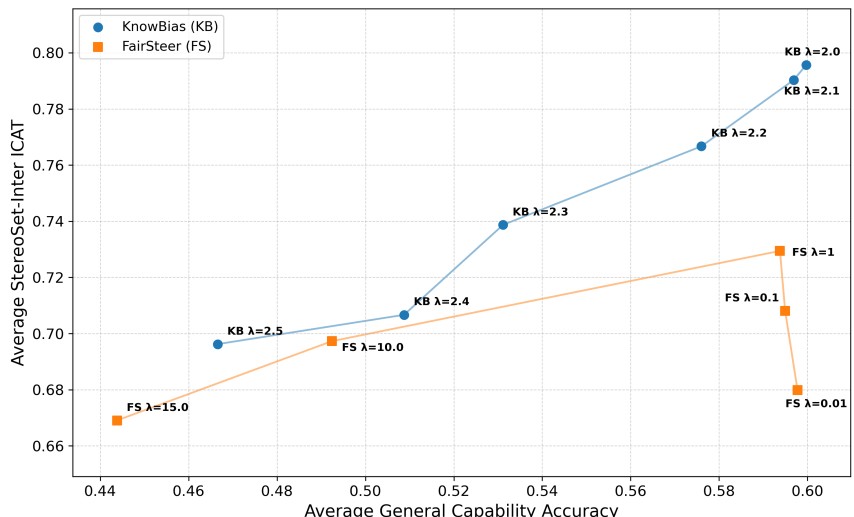

*Figure 4.* Compact fairness-utility tradeoff for KnowBias and FairSteer. Due to the limited rebuttal time window, we include FairSteer as the baseline comparison. Fairness is summarized as the average StereoSet-Inter ICAT score across gender, race, and religion, while utility is summarized as the average accuracy over COPA, OpenBookQA, ARC-C, and ARC-E. Each point corresponds to a different hyperparameter setting for each method: FS $\lambda$ denotes the FairSteer training regularization parameter, and KB $\lambda$ denotes the KnowBias neuron enhancement scale. The selected KnowBias operating point lies in a favorable fairness-utility region.

*Table 17.* The number of know-bias neurons enhanced. We report the number of know-bias neurons enhanced for each model. The values in parentheses are the proportion of know-bias neurons in the entire language model.

| Model | N. of know-bias neurons (% of total neurons) |
|---|---|
| Llama-3.2-3B | 384 (0.17%) |
| Qwen-3-4B | 723 (0.21%) |
| Llama-3.1-8B | 138 (0.03%) |

*Table 18.* Ablation study on bias-knowledge question types across datasets (Llama-3.2-3B-Instruct). Best and second-best results are highlighted in **bold** and underlined, respectively (**BBQ**(|↓|), **CS(0.5)**, and **SS(↑)**).

| Method | Gender | | | | | Race | | | | | Religion | | | | |
|---|---|---|---|---|---|---|---|---|---|---|---|---|---|---|---|
| | BBQ-a | BBQ-d | CS | SS-inter | SS-intra | BBQ-a | BBQ-d | CS | SS-inter | SS-intra | BBQ-a | BBQ-d | CS | SS-inter | SS-intra |
| Type1 | **.0006** | -.6024 | .6759 | .8344 | .7651 | -.0269 | -.6602 | .6385 | .7625 | .5649 | .0245 | -.7168 | .6825 | .6856 | .5937 |
| Type2 | -.0021 | -.6148 | .7221 | .7607 | .7341 | -.0383 | -.6693 | .6886 | .7629 | .5439 | .0361 | -.7461 | .7161 | **.7331** | .5726 |
| Type3 | -.0031 | -.5802 | .7327 | .8299 | .7673 | -.0307 | -.6540 | .6579 | **.8363** | .5672 | .0307 | -.6875 | .6813 | .6629 | .5771 |
| KnowBias | -.0018 | **-.5499** | **.6674** | **.8588** | **.7892** | **-.0257** | **-.6102** | .6316 | .8081 | **.5801** | .0211 | **-.6667** | .6519 | .7202 | **.6031** |

salient sets of at least $\beta\%$ of questions to be selected; and (4) the enhancement scale $\lambda$, which controls the magnitude of activation scaling applied at inference time. This study assesses the sensitivity of KnowBias to these hyperparameters and identifies stable regimes in which debiasing remains effective and consistent.

*Table 19.* Ablation study of cross-dimensional aggregation strategies for know-bias neurons (Llama-3.2-3B-Instruct). Best and second-best results are highlighted in **bold** and underlined, respectively (**BBQ**(|↓|), **CS(0.5)**, and **SS(↑)**).

| Method | Gender | | | | | Race | | | | | Religion | | | | |
|---|---|---|---|---|---|---|---|---|---|---|---|---|---|---|---|
| | BBQ-a | BBQ-d | CS | SS-inter | SS-intra | BBQ-a | BBQ-d | CS | SS-inter | SS-intra | BBQ-a | BBQ-d | CS | SS-inter | SS-intra |
| Random | -.0089 | -.6764 | .8444 | .7108 | .7276 | -.0473 | -.7024 | .7831 | .7597 | .5508 | .0432 | -.7321 | .7388 | .6758 | .5256 |
| KnowBias-S | .0019 | -.5614 | .6697 | .8400 | .7782 | -.0275 | **-.5715** | **.6235** | **.8231** | .5757 | .0286 | -.6945 | .6942 | .6842 | **.6153** |
| KnowBias-q-comp | **-.0018** | -.5820 | .6940 | .7948 | .7599 | -.0409 | -.6422 | .6796 | .8059 | .5431 | .0430 | -.7224 | .7198 | .7017 | .5837 |
| KnowBias-comb | .0045 | -.5872 | .6721 | .8080 | .7721 | -.0437 | -.6141 | .6363 | .7901 | .5732 | .0268 | -.6748 | .6978 | .6481 | .5924 |
| KnowBias-∩ | .0041 | -.5837 | .6798 | .7943 | .7623 | -.0390 | -.6591 | .6673 | .8186 | .5466 | .0358 | -.7232 | .7304 | .6417 | .5849 |
| KnowBias-∪ | **-.0018** | **-.5499** | **.6674** | **.8588** | .7892 | **-.0257** | -.6102 | .6316 | .8081 | **.5801** | **.0211** | **-.6667** | **.6519** | .7202 | .6031 |

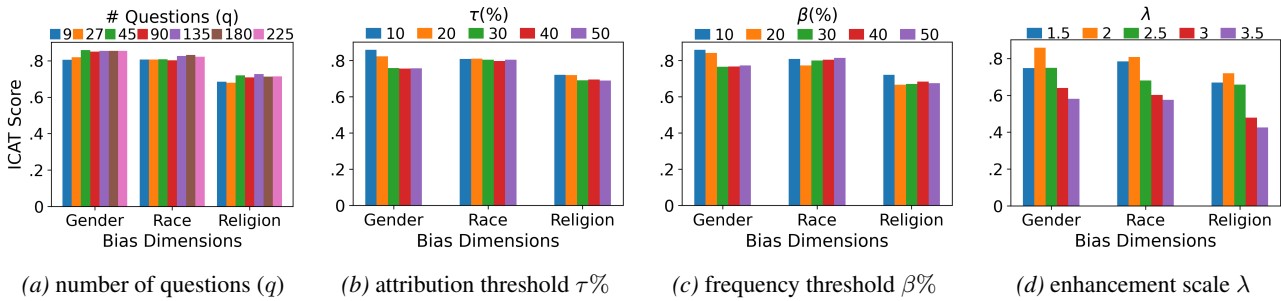

*(a)* number of questions (q)    *(b)* attribution threshold τ%    *(c)* frequency threshold β%    *(d)* enhancement scale λ

*Figure 5.* Hyperparameter study for KnowBias on Llama-3.2-3B-Instruct. ICAT score (↑) for Stereoset-Inter.

### D.8.1. EFFECT OF THE NUMBER OF QUESTIONS q.

We first study the effect of the number of simple bias-knowledge questions $q$ used to identify know-bias neurons. The parameter $q$ controls how many questions are used to identify know-bias neurons: increasing $q$ can potentially improve coverage of bias knowledge, but also increases computational cost and redundancy. To isolate this effect, we fix the other hyperparameters to $\tau = 10\%$, $\beta = 10\%$, and enhancement scale $\lambda = 2$, and vary $q$. As shown in Figure 5a, debiasing performance improves rapidly as $q$ increases from very small values, but saturates once $q$ reaches approximately 45 questions (corresponding to 5 questions per type, 3 question types, and 3 demographic dimensions). Further increasing $q$ beyond this point yields only marginal gains across all three bias dimensions. Thus, we pick $q = 45$ as an optimal value. Importantly, this saturation behavior indicates that KnowBias is **data-efficient**: effective know-bias neurons can be identified using a small number of simply designed bias-knowledge questions, without requiring large concrete demographic identities or stereotype-specific attributes.

### D.8.2. EFFECT OF ATTRIBUTION THRESHOLD τ.

Next, we analyze the sensitivity of KnowBias to the attribution threshold $\tau$, which determines how salient a neuron's attribution score must be (relative to the maximum) to be selected as a candidate know-bias neuron. A lower $\tau$ includes more neurons with moderate attribution, while a higher $\tau$ focuses only on the most salient neurons but risks discarding useful signals. In this experiment, we fix the number of questions to $q = 45$, $\beta = 10\%$, and $\lambda = 2$, and vary $\tau$ from 10% to 50%. The results in Figure 5b show that $\tau = 10\%$ consistently achieves the strongest debiasing performance across gender, race, and religion. As $\tau$ increases, performance gradually degrades, suggesting that overly restrictive thresholds exclude many informative know-bias neurons. This trend supports the use of a relatively low attribution threshold to capture a broader set of relevant neurons.

### D.8.3. EFFECT OF CROSS-QUESTION FREQUENCY THRESHOLD β.

We analyze the cross-question frequency threshold $\beta$, which retains a neuron only if it exceeds the attribution threshold in at least a fraction $\beta$ of bias-knowledge questions. This parameter controls how strictly we enforce cross-question agreement: a larger $\beta$ favors neurons that are repeatedly salient across many questions, but can exclude neurons that contribute to bias knowledge more selectively. We fix $q = 45$, $\tau = 10\%$, and $\lambda = 2$, and vary $\beta$ from 10% to 50%. Figure 5c shows that a loose frequency constraint (e.g., $\beta = 10\%$) produces the most effective neuron set in our experiments, while increasing $\beta$ tends to reduce debiasing performance. These results indicate that bias-knowledge signals are distributed across a diverse

set of neurons, and that overly strict frequency filtering can discard complementary neurons that are useful for debiasing under different question instantiations.

### D.8.4. EFFECT OF ENHANCEMENT SCALE $\lambda$.

Finally, we examine the effect of the enhancement scale $\lambda$, which controls how strongly the activations of know-bias neurons are amplified during inference. A larger $\lambda$ increases the influence of bias knowledge, but excessive scaling may distort the model's internal representations and harm stability. In this experiment, we fix $q = 45$, $\tau = 10\%$, and $\beta = 10\%$, and vary $\lambda$. The results (Figure 5d) show that $\lambda = 2$ consistently yields the best debiasing performance across all three demographic dimensions. Smaller values under-amplify know-bias neurons and lead to weaker effects, while larger values provide no additional benefit and can slightly degrade performance. This finding indicates that a moderate enhancement strength is sufficient to guide model behavior without over-intervening.

### D.8.5. SUMMARY

Overall, the hyperparameter study demonstrates that KnowBias is robust across a wide range of settings and operates effectively in a stable regime with a small number of abstract questions, a low attribution threshold, a loose consistency requirement, and moderate activation enhancement. These results further highlight the practicality of KnowBias, as effective debiasing can be achieved without extensive hyperparameter tuning. And complete hyperparameter settings for all three models are in Appendix D.4.

