# OpenReview forum: "Knowing Bias, Doing Better: Mitigating Social Bias in LLMs via Know-Bias Neuron Enhancement"
_ICML.cc/2026/Conference — ICML 2026 regular_

### Official Review · Reviewer_c5Bx · 2026-02-15

**Soundness:** 3
**Presentation:** 3
**Significance:** 3
**Originality:** 3
**Overall Recommendation:** 4
**Confidence:** 4

**Summary:**

This paper propose KnowBias, a lightweight and conceptually distinct framework that mitigates bias by strengthening neurons encoding bias-knowledge. KnowBias identifies neurons encoding bias knowledge using a small set of bias-knowledge questions via attribution-based analysis, and selectively enhances them at inference time. Experiments across multiple benchmarks and LLMs demonstrate consistent state-of-the-art debiasing performance with minimal utility degradation.

**Compliance With Llm Reviewing Policy:**

Affirmed.

**Final Justification:**

My question has been fully addressed; however, after reviewing the comments from n7Gq, I have decided to retain rathor than improve my original score.

**Key Questions For Authors:**

1. Can you provide more experiments utilizing Gemma-3?
2. Can you provide the statistical significance, confidence intervals, or per-run variance for Table 2 and Table 3?

**Limitations:**

yes

**Strengths And Weaknesses:**

Strengthness:
1. By enhancing know-bias neurons, this method achieves strong generalization to unseen bias types and demographics.
2. The idea that debiasing by strengthening know-bias neurons rather than weakening the bias of neurons is novel.
3. The authors selected a wide range of methods as baselines for comparison, and the experiments are sufficiently comprehensive.
4. This method achieves an excellent balance between debiasing and the general capabilities of large language models.

Weakness:
1.  Llama-3.2-3B-Instruct and Llama-3.1-8B-Instruct employed for experiments is published relatively earlier, you should add more LLMs such as Gemma-3 for experiments.
2. Although the empirical results are broad, the paper is weak on reporting statistical significance, confidence intervals, or per-run variance for its main tables—especially Table 2 and Table 3.

---

> ### Author Rebuttal · Authors · 2026-03-31
>
> We sincerely appreciate the reviewer's supportive evaluation and thoughtful comments. It is encouraging to see that the reviewer finds the proposed idea meaningful, and we thank them for the suggestions that help us further improve the paper.
> ## W1/Q1:
> Thank you for this helpful suggestion.
> - We conduct additional experiments on Gemma-3-4B, and the results are provided at **https://anonymous.4open.science/r/ICML26-18A5/Gemma3.md**.
> - We also add additional experiments on Llama-3.2-3B, including HolisticBias evaluation and Difference Awareness on N1-BBQ, results are at **https://anonymous.4open.science/r/ICML26-18A5/add_eval.md**.
>
> Due to the limited rebuttal time window, we include a reduced set of representative baselines in these additional experiments, while keeping the comparison as informative as possible. These results further support that the effectiveness of KnowBias is not limited to the original backbone family in the main paper.
> ## W2/Q2:
> We also thank the reviewer for this valuable suggestion. To address this concern, we added confidence intervals for the experimental results, which are available at **https://anonymous.4open.science/r/ICML26-18A5/CI.md**.

---

> > ### Author Rebuttal · Reviewer_c5Bx · 2026-04-01
> >
> > My question has been fully addressed; however, after reviewing the comments from n7Gq, I have decided to retain rathor than improve my original score.

---

> > > ### Author Response · Authors · 2026-04-04
> > >
> > > We thank Reviewer c5Bx for the thoughtful follow-up and are glad that our response fully addressed your original questions.
> > >
> > > We understand that you chose to retain your score after considering Reviewer n7Gq's concerns. For completeness, we would like to briefly note that we have also addressed those concerns in the rebuttal:
> > > - Broader model coverage. We added additional experiments on Gemma and more Llama settings.
> > > - Statistical reliability. We added confidence intervals for the main results.
> > > - Evaluation protocol. We clarified that our setup follows standard prior work: each benchmark is evaluated with its standard metric.
> > > - Question set vs. evaluation set. We clarified that the 45 bias-knowledge questions are used only for neuron identification, while debiasing is evaluated on the full benchmark splits.
> > > - Method clarity. We clarified the question template design, the role of abstract bias concepts, and additional details on $\lambda$ fairness–utility tradeoff.
> > >
> > > We believe the main concerns raised by Reviewer n7Gq have now been either addressed with additional evidence or clarified in scope. If there are specific aspects that the reviewer feels would benefit from further development, we would sincerely welcome that guidance. Your comments and follow-up have helped us improve the paper, and we sincerely appreciate that contribution.

---

### Official Review · Reviewer_n7Gq · 2026-03-09

**Soundness:** 2
**Presentation:** 3
**Significance:** 2
**Originality:** 3
**Overall Recommendation:** 3
**Confidence:** 4

**Summary:**

This work introduces KnowBias, a new framework that reduces social bias in large language models by focusing on what the model already knows. This method strengthens the internal neurons that recognize bias instead of trying to hide biased behavior. The process starts by asking the model 45 simple “yes/no” questions to see if it can identify biased opinions. The paper then uses Integrated Gradients to find the specific neurons that help the model answer these questions correctly. At the time of use, KnowBias increases the activity of these neurons to make the model fairer without any training. Tests on benchmarks (e.g., BBQ and StereoSet) show that this approach is very effective and does not hurt the model’s overall reasoning skills.

**Compliance With Llm Reviewing Policy:**

Affirmed.

**Final Justification:**

The rebuttal addressed some of my concerns. It is good that they have added at least two more models and added the confidence interval. However, some other concerns are not addressed properly. For instance, for W2, one metric per dataset is not sufficient to make a general conclusion and the scope is very limitted. The rebuttal tries to justify that they do not claim 'a universal solution for every possible bias metric'.  We know that all fairness metrics are not concurrently achievable. The purpose of testing on multiple data-metrics-models is to see to what extent the method works and where it fails. Regardless. Due to the effort made in the rebuttal period and providing extra results on two more models and the confidence interval, I would like to raise my mark.

**Key Questions For Authors:**

Q1: How does the method work if there are guardrails? For example, if there are hidden prompts asking the model not to generate biased outcomes, then the internal neuron may amplify bias, but the response of the LLM is not biased. How can your method fix the original bias in this setting?

Q2: Do we need to know the concept of bias to find the neuron? In practice we do not have that and we may loose generalizability?

Q3: Each set of questions (e. 15) to find the neurons are mix of different concepts or it is based on one concept only

Q4: For models with significantly larger parameter counts (e.g., Llama 3 70B), does the proportion of identified KnowBias neurons decrease further? Additionally, does the enhancement scale λ require substantial adjustment?

Q5: How sensitive is the neuron identification process to the choice of the baseline value used in the Integrated Gradients calculation? How does this choice impact the stability and consistency of the final recovered neuron set?

**Limitations:**

The limitation is not comprehensive. In my view, making a large claim of debiasing based on 3 dataset and one metric per dataset is a big limitation. Testing the approach on 2 LLM types (3, considering different sizes) and similar parameter sizes is also far from a comprehensive evaluation. Not having a confidence interval also means the results have not been statistically supported.

**Strengths And Weaknesses:**

Strength

Soundness: The ablation study thoroughly examined problem types, aggregation strategies (Union vs. Intersection), and hyperparameter sensitivity, presenting a reasonable argument.

Significance:
Very high data efficiency (only 45 questions) and low computation cost (about 30 mins) to make great performance, significantly outperforming benchmark methods requiring hours of fine-tuning.

Originality: The proposed idea of “enhancing knowledge” rather than “suppressing behavior” is novel and provides a fresh perspective.

Presentation: This paper is clearly structured, with Algorithm 1 and Figure 2 providing an intuitive representation of the workflow.



Weakness
1- The method has been tested on 3 datasets only and two types of models (Llama 3B and 8B and Qwen 3B). The paper used one bias metric for each dataset only. The sensitive attributes are also limited to 3.

2- The bias metrics are very limited, one per dataset, while the claim of dibiasing is a big claim. We expect to see the method work across multiple metrics, models and datasets. The claim that bias metrics are not comparable across datasets is not accurate, and other papers use them across datasets by proper preprocessing.

3- Even in such a limited setting KnowBias method does not consistently perform better than other methods.  Also, in the absence of a confidence interval, it is not possible to judge if the numbers are cherry-picked or random. I suggest adding a confidence interval and reposting the results. Particularly as it took KnowBias less than 30 minutes to extract all KnowBias neurons for 45 questions total. So it is possible timewise.

4- The size of the dataset and evaluation is missing. There are some numbers about the number of questions. I am wondering if you applied this number of questions to find the neuron and evaluate all the BBQ, CS and SS datasets?

5- Table 6 in its current shape is just a list of bias concepts for which you do not need a Table to show. I suggest you add a sample of questions per concept and also present if you have used any specific template to generate questions per concepts.

Minor suggestion:

For  Table 3, it is better to make the model that does not drop the performance bold to be consistent with the common presentation style of other papers, which makes the positive outcome bold.

Overall, given the interesting original idea, the paper will be in good shape for top-tier conferences in the future if it addresses the weakness and makes the analysis comprehensive across models/metrics/datasets/demographics. In its current shape, it is an initial proof of concept.

---

> ### Author Rebuttal · Authors · 2026-03-31
>
> We thank the reviewer for the thoughtful feedback.
> # Weaknesses
> ## W1:
> - The current evaluation is already quite comprehensive compared with prior work: we evaluate across multiple models, multiple social-bias benchmarks, and general-capability tasks, which already provides a broad test of both debiasing performance and utility.
> - To further strengthen the empirical scope, we conducted additional experiments: Gemma (**https://anonymous.4open.science/r/ICML26-18A5/Gemma3.md**) and more Llama results (**https://anonymous.4open.science/r/ICML26-18A5/add_eval.md**).
> ## W2:
> - We respectfully disagree that this weakens our main conclusion. Our goal is not to claim a universal solution for every possible bias metric, but to show that the core KnowBias transfers across standard and widely used benchmarks, model backbones, and demographic dimensions.
> - Even under heterogeneous evaluation protocols, the same overall pattern remains: KnowBias consistently provides strong debiasing with competitive utility preservation.
> ## W3:
> We added confidence intervals for the main results: **https://anonymous.4open.science/r/ICML26-18A5/CI.md**.
> ## W4:
> We would like to clarify that the paper already distinguishes the 45 bias-knowledge questions from the downstream evaluation set: in Section 2.5 and Section 3.4, we state that these 45 questions are used only for neuron identification, while the debiasing performance is evaluated on the full benchmark splits. We agree that this distinction can be made more explicit. The evaluation dataset sizes are in **https://anonymous.4open.science/r/ICML26-18A5/t_s.md**.
> ## W5:
> We agree that Table 6 can be made more informative. The paper already includes sample questions in Table 1 and Figure 2; the bias-knowledge questions are generated from the same reusable template, with different bias concepts instantiated under each demographic dimension. We will revise this part by adding a clearer table of all questions.
> # Questions
> ## Q1:
> - Simply adding extra instructions in the prompt to ask the model not to be biased is often insufficient for debiasing: the model may still exhibit biased behavior despite such test-time prompting.
> - This is not the goal of KnowBias. Our method does not aim to directly suppress or remove biased outputs through external guardrails; instead, it aims to enhance neurons that know bias.
> - If we have misunderstood the reviewer’s intended comparison, we would be very happy to further clarify this point.
> ## Q2:
> No. As stated in Section 2.2, we use only a small set of simple abstract concepts as lightweight probes, and the paper explicitly notes that the goal is not to comprehensively enumerate or carefully tune all possible bias concepts. This is further supported by the data-efficiency experiment in Section 3.4 / Figure 3, which shows that performance already saturates at 45 questions, indicating that KnowBias does not rely on exhaustive concept knowledge to generalize.
> ## Q3:
> As stated in Section 2.5: 15 questions per demographic dimension = 5 concepts × 3 question types.
> ## Q4:
> - Sorry, we may not fully understand the first part of the question regarding the proportion of neurons; if the reviewer could further clarify the intended comparison, we would be happy to address it further. We do our best to answer this question:
>   - We do not assume a fixed neuron proportion across models.
>     - The neuron set of KnowBias is identified adaptively for each model; The number/proportion of identified neurons can naturally vary across models, rather than being fixed in advance.
>   - We have not tried to infer a universal law.
>     - Our current work does not make a claim that the proportion of KnowBias neurons should remain constant or monotonically decrease as model size grows.
>     - How neuron proportions change across scales is an interesting empirical question and worth furture study, but we do not attempt to summarize a unified pattern in this work.
> - $\lambda$ does not require substantial adjustment.
>   - In Appendix D.7, performance is relatively stable over a reasonable range of $\lambda$, while overly large $\lambda$ can hurt general ability.
>   - We additionally include an experiment reporting the fairness-utility curve over different $\lambda$ values: **https://anonymous.4open.science/r/ICML26-18A5/pareto_.md**.
> ## Q5:
> - We would first like to clarify that neuron identification itself is an active research area, and in this work we follow the standard attribution-based setup used in prior knowledge neuron studies rather than proposing a new neuron method.
> - Our existing results already suggest that the recovered KnowBias neuron set is stable enough to support consistent debiasing: the data-efficiency study in Section 3.4 / Figure 3 (Appendix D.7.1 / Figure 4a) shows that performance improves with more bias-knowledge questions and then saturates around 45 questions, indicating that the identified neuron set is not highly fragile to the exact question sample.

---

> > ### Author Rebuttal · Reviewer_n7Gq · 2026-04-01
> >
> > The rebuttal addressed some of my concerns. It is good that they have added at least two more models and added the confidence interval. However, some other concerns are not addressed properly. For instance, for W2, one metric per dataset is not sufficient to make a general conclusion and the scope is limitted. The rebuttal tries to justify that they do not claim 'a universal solution for every possible bias metric'.  We know that all fairness metrics are not concurrently achievable. The purpose of testing on multiple data-metrics-models is to see to what extent the method works and where it fails.
> >
> > Regardless. Due to the effort made in the rebuttal period and providing extra results on two more models and the confidence interval, I would like to raise my mark.

---

> > > ### Author Response · Authors · 2026-04-04
> > >
> > > We thank Reviewer n7Gq for the thoughtful follow-up and continued engagement with our work. We agree that broader evaluation across datasets, models, and metrics is valuable for understanding both where a debiasing method works and where it may fail.
> > >
> > > At the same time, we would like to clarify the scope of our claims and why we believe **the current evaluation remains valid**:
> > > - We do not claim completeness across all fairness metrics or settings. Instead, our claim is that KnowBias achieves consistent gains under the standard primary metric of each widely used benchmark, while maintaining competitive utility across models and tasks.
> > > - We agree that a broader multi-metric analysis would be valuable and is worth future study. However, to our knowledge, there is currently no single widely accepted unified metric for social bias evaluation across these benchmarks; [3] instead treats metrics as benchmark-specific and discusses multiple fairness notions rather than one universally adopted score.
> > > - Accordingly, we follow the standard benchmark-specific evaluation protocol used in prior debiasing works. For example, BiasEdit[1] and FairSteer[2] also evaluate each benchmark with its standard target metric, which is the same evaluation principle we adopt here.
> > > - We believe the current setup provides a fair and valid basis for evaluating KnowBias and comparing it against prior methods.
> > >
> > > Therefore, we believe the current evaluation is still methodologically valid and appropriate for supporting the specific claims we make, especially given that it already spans multiple backbones, datasets, demographic dimensions, and utility benchmarks.
> > >
> > > [1] BiasEdit: Debiasing Stereotyped Language Models via Model Editing. TrustNLP 2025.
> > > [2] FairSteer: Inference Time Debiasing for LLMs with Dynamic Activation Steering. ACL 2025.
> > > [3] Bias and Fairness in Large Language Models: A Survey. Computational Linguistics 2024.

---

### Official Review · Reviewer_ntDq · 2026-03-12

**Soundness:** 4
**Presentation:** 3
**Significance:** 4
**Originality:** 3
**Overall Recommendation:** 5
**Confidence:** 5

**Summary:**

This work identifies and mitigates biases in three steps: (1) elicit bias using a dataset containing questions along three concepts (causal rejection, bias recognition, normative judgment), (2) use a neuron attribution method to identify which neurons contribute most to bias, and (3) strengthen the identified and aggregated neurons (across various demographics, selecting those that activate for at least beta% of questions) using a lambda multiplier to mitigate bias. They evaluate on multiple bias datasets (BBQ, CrowsPairs, StereoSet) and call this framework KnowBias. KnowBias is shown to, for the most part, consistently outperform other baselines (prompt-based, edit-based, steering, and fine-tuning). They also show that (1) only a small number of samples (45) are required to see benefits in bias mitigation, and (2) KnowBias generalizes to out-of-distribution demographics on the StereoSet dataset.

**Compliance With Llm Reviewing Policy:**

Affirmed.

**Key Questions For Authors:**

1. In the prompt-based baseline adopted from Gallegos et al., what was the reprompting strategy used: explanation-based or reprompting-based?

2. What is the computational cost of neuron attribution at scale? For larger models, how does the per-neuron computation scale, and is this practical?

**Limitations:**

Yes.

**Strengths And Weaknesses:**

Strengths:
1. The idea of identifying neurons that encode bias knowledge and amplifying them to mitigate bias is intuitive and simple.
2. The experimental evidence largely supports the authors' claims about the merits of KnowBias. Data efficiency and generalization in particular are evident from the results.
3. Ablations are complete: on number of questions, lambda, beta, and tau.
4. Baseline coverage is good, especially conceptually (prompt-based, edit-based, steering, fine-tuning).
5. Largely easy to follow, and results are presented adequately.
6. Construction of the bias-eliciting dataset is simple, intuitive, well situated (in the framing of causal rejection, bias recognition, and normative judgment), and easily scalable even if not already complete.

Weaknesses:
1. Model coverage could be better. The paper only studies the Llama and Qwen families. Results on a broader set of architectures (such as Deepseek) would strengthen the claims.
2. There is significant repetition in describing the three features of KnowBias: once in the intro (lines 88-107), again on the same page (Section 2.1, paragraph 2), again in Section 2.5 (summarizing features before experimental results are even presented), and once more rephrased in Section 3 (paragraph 1). I would strongly consider deduplicating, as the repetitions do not add additional value. The space saved could be used to move related work into the main body.
3. The paper could use a clearer formalization of bias beyond the individual definitions operationalized by the metrics used.
4. The prompt-based debiasing baseline adopted is zero-shot, but it is well established that few-shot debiasing is considerably more effective. This weakens the comparison.
5. A hidden cost that is not discussed: neuron-level contribution towards bias needs to be computed on a per-neuron basis, which could be expensive for large models. Some discussion of scalability here would be helpful.
6. I don't believe rank is a good metric to capture fragility of other methods, as it ignores the magnitude of change in bias between models and reduces improvements to relative ordering.

---

> ### Author Rebuttal · Authors · 2026-03-31
>
> We thank the reviewer for the encouraging evaluation, constructive comments, and helpful suggestions. We are glad that the reviewer finds the core idea meaningful, and we address each point below concisely.
> # Weaknesses
> ## W1 model coverage:
> - We thank the reviewer for this suggestion and agree that broader backbone coverage would further strengthen the paper.
> - To address this point, we conducted additional experiments on Gemma-3-4B (additional results: **https://anonymous.4open.science/r/ICML26-18A5/Gemma3.md**). These results further support that the effectiveness of KnowBias is not limited to the original model family evaluated in the main paper.
> ## W2 repetition in presenting:
> Thank you for pointing this out. We agree and will fix this in the revision by removing redundant descriptions and improving the overall presentation.
> ## W3 clearer formalization of bias:
> Thank you for this helpful suggestion. Our current paper defines social bias in terms of harmful stereotypical or unjust social associations (Section 1), while different benchmarks instantiate this notion through their standard evaluation protocols. We will clarify this more clearly in the revision.
> ## W4 few-shot evaluation:
> Thank you for this suggestion. Our goal in this work is generalizable debiasing, where the intervention can be applied without using any task-specific testing information. For this reason, our experiments focus on the zero-shot setting. There may be a misunderstanding on our side regarding the exact few-shot setting the reviewer has in mind: if the reviewer refers to using task-specific in-context examples at test time, this would be less aligned with our goal of studying debiasing as a general intervention independent of the downstream test instances. We would be happy to further discuss this point if the reviewer has a different few-shot setting in mind.
> ## W5 scalability / computational cost:
> - The scalability/cost issue mainly comes from the current neuron-identification methodology, which is part of a broader and still active research area. The bottleneck is not the KnowBias intervention itself, but the cost of identifying neuron sets using existing neuron-discovery tools.
> - Our contribution is the KnowBias framework to mitigate social bias, which is independent of any specific neuron-finding tool.
> - The attribution-based method we use is one reasonable instantiation, but it can in principle be replaced by any improved or more scalable neuron discovery method.
> - In our method, this cost comes from the neuron identification step, while the debiasing framework itself is lightweight once the neuron set is identified.
> - We will add a clearer discussion of this cost and scalability trade-off in the revision.
> ## W6 ranks usage:
> - Thank you for this helpful comment. The rank-based comparison does not fully reflect the magnitude of the differences between methods. At the same time, our motivation for using ranks is that the evaluated benchmarks and metrics are on different scales and are therefore not directly comparable for simple averaging across datasets.
> - We used ranks as a normalized summary to provide an aggregate view across heterogeneous evaluations. We will clarify this motivation more explicitly in the revision.
>
> # Questions
> ## Q1:
> - Thank you for this question. We use self-debiasing reprompting because our goal is to evaluate generalizable debiasing, i.e., a method that can be applied uniformly across arbitrary downstream tasks without assuming task-specific formats, extra reasoning steps, or additional test-time information.
> - In contrast, reexplanation requires the model to generate an explicit bias explanation before answering, which changes the original inference procedure and is therefore less suitable for evaluating general downstream tasks and general ability benchmarks.
> ## Q2:
> See W5 scalability / computational cost.

---

> > ### Author Rebuttal · Reviewer_ntDq · 2026-03-31
> >
> > My concerns were adequately addressed. I will raise my score.

---

> > > ### Author Response · Authors · 2026-04-04
> > >
> > > We thank Reviewer ntDq for the positive evaluation and continued constructive engagement with our work. We are grateful that the reviewer recognized both the novelty of the idea and the overall strength of the paper, and we appreciate the detailed suggestions for improving clarity and presentation. The feedback has been very helpful in further strengthening the paper.

---

### Official Review · Reviewer_51xf · 2026-03-16

**Soundness:** 2
**Presentation:** 4
**Significance:** 3
**Originality:** 3
**Overall Recommendation:** 5
**Confidence:** 3

**Summary:**

The authors tackle the problem of debiasing LLMs for social/demographic biases. They propose KnowBias, an inference-time method that first identifies neurons which encode LLM's acknowledgement of bias, and then amplifies these activations during generation. Across evaluations on five bias benchmarks and three base LLMs, they find that KnowBias reduces bias scores across three demographic groups, while preserving general capabilities on four reasoning datasets.

**Compliance With Llm Reviewing Policy:**

Affirmed.

**Final Justification:**

The authors have sufficiently addressed my concerns in their rebuttal by providing additional experiments. I have raised my score to a 5.

**Key Questions For Authors:**

Please address the weaknesses above, and the following questions:

1. Why do the authors think the method works so much better than CRISP, which has a similar idea? How similar are the overlap of the neurons identified by both method (we would expect them to be disjoint)? Is it possible to combine these two methods?

2. Why are BBQ-d scores so poor in general?

**Limitations:**

yes

**Strengths And Weaknesses:**

Strengths:
- The paper is well-written and easy to understand.
- The proposed method makes sense conceptually, and the authors draw interesting motivation from cognitive science.
- The method is practical (no gradients required at inference time).

Weaknesses:

1. The authors choose $\lambda$ in an ad-hoc way for each of the models, and report their main result tables (Tables 2 and 3) only for a single value of $\lambda$. Instead, a much better evaluation would be what is typical in the fairness literature, which is to plot the pareto front of fairness (i.e. a summarized Table 2 number) versus overall performance (a summarized Table 3 number), varying $\lambda$, and also varying appropriate hyperparameters for baseline methods to create a curve (if applicable). The goal is to have KnowBias Pareto-dominate the baselines. This would allow a reasonable comparison with e.g. FairSteer and LFTF, which do better than KnowBias on general reasoning but worse on fairness.

2. The authors should also evaluate methods in contexts where it is necessary and desirable to discriminate across the demographic group (see [1]), to make sure that KnowBias does not degrade performance in these cases.

3. All the bias evaluation datasets are MCQA or probability-based evaluations, instead of open-ended generation which is more practical. Some datasets which the authors should test on are BOLD [2] and HolisticBias [3].

4. The authors should test on some slightly larger LLMs if compute allows, as larger LLMs generally tend to exhibit less bias [4].


[1] Fairness through Difference Awareness: Measuring Desired Group Discrimination in LLMs. ACL 2025.

[2] BOLD: Dataset and Metrics for Measuring Biases in Open-Ended Language Generation. FAccT 2021.

[3] "I'm sorry to hear that": Finding New Biases in Language Models with a Holistic Descriptor Dataset. EMNLP 2022.

[4] The Capacity for Moral Self-Correction in Large Language Models. arXiv:2302.07459.

---

> ### Author Rebuttal · Authors · 2026-03-31
>
> We thank the reviewer for the constructive feedback and helpful suggestions. We are encouraged that the reviewer finds the core idea promising. Below we address each point concisely.
> # Weaknesses
> ## W1 $\lambda$ selection:
> We agree that a Pareto-style view is valuable.
> - Our choice of $\lambda$ is not ad hoc: Appendix D.7.4 already includes a sensitivity study of the bias score of Stereoset-inter across a range of $\lambda$.
> - To address the reviewer more directly, we additionally ran general ability accuracy under different $\lambda$ and show the resulting fairness–utility curve at: **https://anonymous.4open.science/r/ICML26-18A5/pareto_.md**. This makes the trade-off explicit and shows that KnowBias remains competitive in utility while improving fairness.
> ## W2 & W3 additional experiments (Difference Awareness and Holistic Bias):
> We thank the reviewer for this helpful suggestion. We agree that, beyond controlled bias benchmarks, it is important to examine whether the method preserves legitimate demographic distinctions and generalizes to open-ended generation.
> - To address this, we conducted additional experiments on Llama-3.2-3B, including HolisticBias evaluation and Difference Awareness on N1-BBQ.
> - The additional results are available at **https://anonymous.4open.science/r/ICML26-18A5/add_eval.md**. Due to the limited rebuttal time window, we include a reduced set of representative baselines in these additional experiments, while keeping the comparison as informative as possible.
> ## W4 evaluation on larger LLMs:
> - We agree that testing on larger LLMs would further strengthen the paper. At the same time, due to computation constraints, our experiments focus on model scales that allow fair and consistent comparison across all methods.
> - Several baselines in our study also require nontrivial tuning or intervention-specific optimization, and the existing debiasing literature in our comparison does not typically evaluate on substantially larger models either.
> - Our goal in this work is to study debiasing under a fair experimental setup, where all methods are compared under comparable model scales and resource conditions. From this perspective, we believe our current evaluation is a fair testbed for assessing debiasing effectiveness, and we hope the reviewer can understand this practical trade-off.
>
> # Questions
> ## Q1:
> Although both methods operate on neurons, they are fundamentally different in what they identify and how they intervene:
> - CRISPR identifies neurons associated with biased behavior (bias neurons); KnowBias identifies neurons that know bias (know-bias neurons).
> - On neuron difference: we show the identified neurons for both CRISPR and KnowBias on Llama3.2-3B at https://anonymous.4open.science/r/ICML26-18A5/neurons.md; the two methods recover clearly different neuron sets, consistent with their different mechanisms.
> - CRISPR aims to mitigate bias by removing bias neurons; KnowBias instead enhances know-bias neurons to debias.
> - Effect on general capability: empirically, under comparable debiasing effectiveness, KnowBias better preserves common general abilities than CRISPR.
> - We believe this distinction is important because enhancing know-bias neurons is inherently less destructive than removing neurons that may also support other useful behaviors.
> - We also agree that the two approaches could potentially be combined: since CRISPR and KnowBias act on different neuron sets with different mechanisms, they are likely complementary. At the same time, such a hybrid design would need to be carefully balanced, since neuron elimination can further reduce biased behavior but overly aggressive suppression may also degrade general ability. Therefore, combining mild elimination with know-bias enhancement is a promising but out-of-scope direction for the current paper, which we will briefly note as future work in the revision.
> ## Q2:
> - We thank the reviewer for raising this point. We do not believe this is specific to KnowBias; rather, BBQ-d is a particularly challenging setting under zero-shot evaluation.
> - Compared with more direct bias-measurement settings, BBQ-d requires the model to simultaneously follow contextual evidence and avoid stereotypical reasoning, which makes the task substantially harder.

---

> > ### Author Rebuttal · Reviewer_51xf · 2026-04-03
> >
> > Thank you for the response. My concerns have been addressed, and I will raise my score to a 5.

---

> > > ### Author Response · Authors · 2026-04-04
> > >
> > > We thank Reviewer 51xf for the thoughtful and encouraging feedback on both the paper and the rebuttal. We especially appreciate the reviewer's recognition of the core idea and the constructive suggestions for further strengthening the work. The comments have helped us improve the paper’s evaluation and overall presentation.

---

### Decision · Program_Chairs · 2026-04-30

**Decision:**

Accept (regular)

**Comment:**

This paper received two Accept, one Weak Accept and one Weak Reject final recommendations.

On the one hand, all the reviewers appreciate the novelty of the proposed approach, based on strengthening, rather than suppressing, neurons that encode bias knowledge, while recognizing the paper as well-written, clearly structured and easy to follow. Furthermore, the efficiency of the KnowBias method, the overall supporting evidence, with the inclusion of appropriate baselines, are recognized as strengths.

On the other hand, reviewers raised concerns regarding the effectiveness of the method for open-ended generation and more architectures, requiring the extension of experiments to cover appropriate datasets and models, the necessity to expand the analysis to more sensitive attributes and the usage of a single bias metric.

In their rebuttal, the authors provided additional clarification and experiments, covering more models and datasets, and providing confidence intervals.

All the reviewers generally acknowledged and appreciated the details and specifications provided in the rebuttal. Most of the initial concerns are considered resolved, however, doubts persist, especially regarding the usage of a single bias metric per dataset.

After reading the paper, and carefully evaluating the initial reviews, rebuttal, and discussion, the AC shares the identified strengths, especially in terms of novelty of the approach and the clarity of the paper and values the added experiments.

Therefore, while sharing some of the remaining doubts, considering the strengths of this work and the overall recommendations, the AC recommends acceptance, given that the additional clarifications and experiments are incorporated in the camera-ready version of the manuscript.